# Comparative efficacy and safety of the artemisinin derivatives compared to quinine for treating severe malaria in children and adults: A systematic update of literature and network meta-analysis

**Nicholas Nyaaba**[1]*, **Nana Efua Andoh**[2◉], **Gordon Amoh**[3◉], **Dominic Selorm Yao Amuzu**[4◉], **Mary Ansong**[5◉], **José M. Ordóñez-Mena**[6,7‡], **Jennifer Hirst**[6,7‡]

**1** Infectious Disease Centre, 37 Military Hospital, Cantonments, Accra, Ghana, **2** Noguchi Memorial Institute for Medical Research, University of Ghana, Accra, Ghana, **3** Korle-Bu Polyclinic/ Family Medicine Department, Korle-Bu Teaching Hospital, Korle-Bu, Accra, Ghana, **4** West African Centre for Cell Biology of Infectious Pathogens, Department of Biochemistry, Cell and Molecular Biology, University of Ghana, Legon, Accra, Ghana, **5** The International Sickle Cell Centre, Accra Central, Accra, Ghana, **6** Nuffield Department of Primary Care Health Sciences, University of Oxford, Oxford, United Kingdom, **7** National Institute for Health Research (NIRH), Oxford Biomedical Research Centre, Oxford University Hospitals NHS Foundation Trust, Oxford, United Kingdom

◉ These authors contributed equally to this work.
‡ JOM and JH are Joint Senior Authors.
* nyabasmail@gmail.com

**Data Availability Statement:** This study did not directly receive any funding. Initial part of this

## Abstract

### Background

The artemisinin derivatives are the preferred antimalaria drugs for treating severe *Plasmodium falciparum* malaria. However, their clinical effectiveness compared to each other is unknown. Our objective, therefore, was to evaluate the efficacy and safety of the artemisinin derivatives and quinine for treating severe *P. falciparum* malaria in children and adults using a network meta-analysis.

### Methods and findings

Review protocol was registered with PROSPERO, CRD42020218190. We updated the search strategies of three Cochrane systematic reviews which included published and unpublished randomised control trials (RCTs) that have compared specific artemisinin derivatives to quinine in treating severe malaria. Search included CENTRAL, MEDLINE, Embase, LILACS, ISI Web of Science and trial registries up to February 2021. We screened studies, extracted data, assessed risk of bias, and quality of evidence in duplicate. Separate network meta-analyses in the frequentist framework, using a random effects model, with quinine as reference, were conducted for adults and children, and rankings were produced using p-scores to assess mortality, parasite clearance, coma recovery, fever clearance, neurological sequela and adverse events.

review was done as part of Nicholas Nyaaba's MSc from the University of Oxford which was supported by the Ghana Education Trust Fund (GETFund) and the Ghana Armed Forces Medical Services (GAFMS). The funders had no role in study design, data collection and analysis, decision to publish, or preparation of the manuscript.

**Funding:** The authors received no funding for this work.

**Competing interests:** The authors have declared that no competing interests exist.

**Abbreviations:** 95%CI, 95% Confidence Intervals; AME, Artemether; AMI, Artemisinin; ATE, Arteether; ASU, Artesunate; IM, Intramuscular; IV, Intravascular; MD, Mean Difference; NMA, Network Meta-analysis; OR, Odds Ratio; QN, Quinine; RCT, Randomised Control Trial; RR, Risk Ratios.

Searches identified 818 citations, 33 RCTs were eligible. We pooled 7795 children and 3182 adults. The networks involved artesunate, artemether, rectal artemisinin, arteether and quinine. Compared to quinine, artesunate reduced mortality in children (risk ratio (RR), 0.76; 95%CI [0.65 to 0.89], moderate quality), adults (RR, 0.55; 95%CI [0.40 to 0.75], moderate quality) and in cerebral malaria (RR, 0.72; 95%CI [0.55 to 0.94], moderate quality).

Compared to rectal artemisinin and intramuscular arteether, the efficacy and safety of parenteral artesunate, and intramuscular artemether in treating severe malaria are not clear. Rankings showed that none of the artemisinin drugs were consistently superior in all the outcomes assessed. Indirect evidence produced were of very low ratings due to suspected publication bias and imprecision.

## Conclusions

Artesunate reduces mortality compared to quinine for both adults and children in Asia and Africa including cerebral malaria. The artemisinin derivatives remain the best treatment for severe malaria but their comparative clinical effectiveness is yet to be fully explored.

## Introduction

In 2019 alone, about 409,000 malaria deaths were recorded among 229 million cases globally, with Africa accounting for 94% of cases and children under 5 years accounting for 67% of all deaths [1]. Although major milestones have been achieved over the last two decades, about half of the world's population is still exposed to malaria, most living in Sub-Saharan Africa, and South East Asia [2]. Malaria is a febrile illness that is spread through the infected bites from female Anopheles mosquitos and it is caused by the *Plasmodium spp* (*P.malariae*, *P.falciparum*, *P.vivax*, *P.knowlesi*, *P.ovale and P.cynomolgi*), of which *P.falciparum* is the major cause of severe illness, therefore, severe *P.falciparum* malaria is the focus of this study [3].

Severe malaria is the serious form of malaria that is fatal and may cause long-term neurological disability. It is diagnosed as positivity to malaria parasite, with life threatening syndromes such as coma, hypoglycaemia, severe anaemia, significant bleeding, convulsions and respiratory distress. Cerebral malaria is the most deadly form of severe malaria and manifest as unarousable coma [4]. Severe malaria is a medical emergency and most deaths occur in the first 48 hours of admission [5]. In addition to supportive treatment, there is a need for prompt administration of an effective antimalarial drug that achieves quick therapeutic plasma concentrations and faster parasite clearance. These antimalarials are administered as parenteral or rectal monotherapies until the patient can tolerate oral formulations [6].

Quinine was the standard treatment for severe malaria for many years but has been associated with treatment resistance and mild to serious adverse events such as tinnitus, deafness, and hypoglycemia. Rapid intravenous infusion can quickly reach toxic levels leading to blindness and death [7].

Since 2011, the artemisinin derivatives have been the preferred antimalarials for treating severe *P.falciparum* malaria [6]. The artemisinin derivatives provide faster clearance of parasites compared to quinine. There is evidence to indicate cardiotoxicity, and neurotoxicity in animal studies for the artemisinin derivatives, but not in humans [7,8]. Artemisinin treatment failure has become the major threat to malaria elimination. Currently, there are reports of

artemisinin resistance originating from South East Asia [9]. This has triggered the need to explore different combination therapies and better use of these drugs [10].

Several systematic reviews of randomised control trials (RCTs) have made comparisons of specific artemisinin derivatives to quinine for treating severe malaria [11–18], providing evidence on their efficacy and safety against quinine. This has led to parenteral artesunate and intramuscular (IM) artemether to be generally accepted as the first and second line treatments respectively, but there is a paucity in the evidence concerning the other artemisinin derivatives for treating severe malaria [14–18].

Currently, no known RCT has compared all the artemisinin derivatives in the same head-to-head study for treating severe malaria. A network meta-analyses (NMA) will therefore provide indirect evidence of the drugs that have never been compared. NMA will also provide rankings for the drugs across relevant outcomes [19].

Our objective was therefore to evaluate the efficacy and safety of the artemisinin derivatives and quinine for treating severe *P. falciparum* malaria in children and adults using an NMA.

## Methods

This systematic review update and NMA was conducted according to the PRISMA guidelines extension [20,21] (S1 Checklist). The NMA protocol was registered with PROSPERO registration number CRD42020218190 [22].

### Interventions

This review covered parenteral and rectal artemisinin drugs administered during the critical phase of treating P. *falciparum* severe malaria until oral antimalarial can be tolerated by the patient. Parenteral interventions cover both IV and IM interventions. Just as IV and IM artesunate are considered to be similar, IV and IM quinine do not have significantly different therapeutic benefits in treating severe malaria [23]. Rectal artemisinin has been compared to parenteral quinine and artesunate in previous studies and has been found to be equally effective in treating severe malaria [24–26]. Therefore, both parenteral and rectal interventions were pooled together in the analyses. However, adverse events were analysed and interpreted with the assumption that local reactions will differ with route of administration. Different dosages for the interventions within recommended ranges were combined into a single node, therefore, trials comparing the same drug at different doses were excluded.

### Inclusion and exclusion criteria

We included all published and unpublished RCTs from three Cochrane systematic reviews that compared artemisinin derivatives to quinine in treating severe malaria [14–16]. In addition, we combined the search strategies of these previous reviews and updated them by searching the Cochrane Infectious Diseases Group Specialized Register; Cochrane Central Register of Controlled Trials (CENTRAL) in the Cochrane Library; MEDLINE, Embase, LILACS, ISI Web of Science and trial registries from inception up to February 2021. Example of search strategy shown in Table A in S1 File. There were no limitations regarding language, geographical setting and year of publication. The criteria for inclusion ensured that all RCTs included in this review compared the artemisinin derivatives and quinine for the treatment of P. *falciparum* severe malaria among adults and children, in head-to-head comparisons. Children were considered as those aged < 15 years. Trials which included pregnant women were excluded.

## Selection of studies

The review team, blinded to each other, independently screened titles and abstracts, and full-text for eligible studies in duplicate, using Rayyan review software [27]. Disagreements were resolved by other co-reviewers.

The primary outcome was proportion of death from all causes compared among drugs from onset of treatment. Secondary outcomes were coma recovery time, parasite clearance time, fever clearance time, neurological sequela events among survivors and adverse events. Hypoglycemia and electrocardiogram abnormalities which were the most frequently reported adverse events were included in the analyses and the others were reported as number of events and proportion experiencing the event. Mortality, neurological sequela, hypoglycemia and electrocardiogram abnormalities were assessed as binary outcomes whilst the rest as continuous outcomes.

## Data extraction

Participant and study characteristics, and outcome data were extracted using a structured data form, in duplicate by NEA, GA, DA, MA and NN. For binary outcomes, the number of participants experiencing the event and numbers assessed in each randomised group were recorded. For continuous outcomes, arithmetic means and standard deviations for each intervention group, with the numbers assessed in each group were extracted. If the number assessed in each group was not reported, the number randomised in the intervention arm was used.

Where medians and range or interquartile range were reported instead of the means and standard deviations, the latter were estimated using Wang's method [28]. These estimations were only used in sensitivity analyses.

Descriptive statistics were presented in Table 2, which outlined individual studies in rows and the study characteristics in columns. These study characteristics included year of publication, sample size, age group, mean age, sex distribution, length of follow up, study continent and intervention arms.

## Assessment of transitivity

The interventions in the study were similar in all comparisons. Also, the participants in the adult and children analyses were similar in all studies and could have been assigned to any of the treatments as indicated above, therefore, meeting the joint randomisability requirement for NMA.

We compared summary characteristics for all included RCTs, across age groups and type of severe malaria (Table 2). All potential effect modifiers' frequency distributions of RCTs were also compared across treatment comparisons (Table B in S1 File). The effect modifiers considered were age group, type of severe malaria, study continent and publication year.

## Assessment of risk of bias

The assessment of risk of bias was done independently in duplicate by NEA, GA, DSYA and MA using the reviewed Risk of Bias Tool (ROB2) [29]. ROB2 assesses risk based on quality of randomisation, whether there were deviations from intended interventions, missingness of outcome data, measurement of the outcome and selection of the reported result. Disagreements were resolved by discussion with the other co-reviewers.

## Statistical analyses

Separate analyses were conducted for adults and children to reduce clinical inconsistency and heterogeneity, as well as meet the transitivity requirement. NMAs were conducted using R

(version 3.6.0) *netmeta* package (version 1.2–1) in a frequentist framework [30]. Risk Ratios (RR) were pooled for mortality, whiles Mean Differences (MD) for coma recovery time, parasite clearance time and fever clearance time, using a random effects inverse variance model [31]. Hypoglycemia events were considered as rare events, therefore, NMA was performed using Mantel Haenszel method [32]. We analysed neurological sequela events and ECG abnormalities with traditional meta-analyses using Peto's method with *metabin* command of the *meta* package because there were two few comparisons and events to conduct an NMA [33]. We however, interpreted these results using RR by converting to Odds Ratios (OR) using the assumed comparator risk which was arbitrarily chosen as the median comparator risk [19]. The NMA estimates were presented in a table with a 95% CI. All statistical tests were two-sided with a significance level of 0.05. Network graphs were created with Stata 15 *network map* [34]. The nodes of the network graph represent the treatments and the edges the comparisons. The bigger the size of the nodes, the greater the pooled sample size in the treatment. The thicker the edges, the greater the number of studies comparing the treatment. League tables and forest plots were used to summarise effect sizes for all possible comparisons, outcomes and subgroups.

## Assessing heterogeneity and inconsistency

The DerSimonian and Laird method was used to estimate the between-study variance, and the Jacksons method to estimate its confidence intervals [31,35]. Variability in effects sizes was assessed globally for the whole network and locally at each possible study design, and described using the Q and $I^2$ statistics, degrees of freedom, and p-values. The variability among the individual study estimates as compared to the network estimates was represented by the total variability, using the Cochran Q method [36]. This was then broken down into within-design (heterogeneity) and between-design (inconsistency). Analyses which recorded substantial inconsistencies were further investigated to identify hotspots using net-splitting and design by treatment methods as appropriate, and illustrated with the net-splitting and net heat plots respectively [36,37].

**Ranking.** Probability scores (p-scores) were used to rank treatments in each outcome. P-scores are based on effect sizes and standard errors only. Forest plots for network results were used to present rankings of each artemisinin relative to quinine alongside p-scores to minimise misinterpretations [38].

**Additional analyses.** Data from adults and children were combined to conduct subgroup analyses by severe malaria type, study continent, and time point of neurological sequela events. In sensitivity analyses, the NMA was repeated for parasite clearance time, coma recovery time and fever clearance time, with the addition of outcome data that were estimated from medians with range or interquartile range instead of means and standard deviations. NMA was also repeated for all age groups combined.

## Assessment of publication bias

In place of the traditional funnel plot that represents each pairwise comparison, a comparison-adjusted funnel plot that incorporates all the effects of publication bias in the network was used, in addition to Egger's test [30].

## Assessment quality of evidence

The quality of evidence was assessed by NEA and NN independently, using the latest version of Confidence in Network Meta-analysis (CINeMA) [39] approach which classifies evidence as high, moderate, low or very low quality based on the within-study bias, heterogeneity,

reporting bias, imprecision, indirectness and incoherence. Disagreements were resolved by discussion with other co-reviewers.

## Results

The search update identified 818 citations. We excluded 134 duplicates and then 641 citations after screening titles and abstracts, 43 full texts were then evaluated. Ten were excluded (see reasons in S1 File), leaving four eligible RCTs [40–43] plus 29 RCTs from the three Cochrane systematic reviews. We pooled 10977 participants; 7795 children and 3182 adults from 33 eligible RCTs as seen in Fig 1.

Summary of study characteristics are presented in Table 1. One RCT was conducted in the South Pacific, 15 in Asia, and 17 in Africa. Nineteen RCTs were conducted in children, 12 in adults and two included both. Sixteen (48%) had sample size greater than 100. Fifteen RCTs reported follow up time of 28 days or more. Twelve RCTs were among participants with only cerebral malaria whiles 21 others were among a mixed population of all severe malaria syndromes. Distribution of study characteristics among subgroups is shown in Table 2. Sixteen out of 17 RCTs from Africa were conducted among children and 11 out of 12 RCTs conducted among adults were in Asia. Out of the 12 cerebral malaria only RCTs, eight were from Africa and four from Asia. Four different artemisinin drugs were compared to quinine for treating severe malaria. Quinine was a comparator in 31 trials. Artemether (n = 22), and artesunate (n = 12) were the most studied artemisinin drugs. The RCTs included eight study designs of which three were multi-arm. The multi-arm designs were artesunate vs artemisinin vs quinine [24,26], artesunate vs artemether vs quinine [42] and IM artesunate vs intravenous (IV) artesunate vs artemisinin vs artemether [25]. Both the IM and IV artesunate arms of the four-armed were combined as one parenteral arm for analyses [25].

Out of these designs, we found seven direct pairwise comparisons (Table B in S1 File). The most studied comparisons were artemether vs quinine (n = 20), and artesunate vs quinine (n = 10). Two comparisons comparing artesunate vs artemether were conducted in Asia among adults. Two trials comparing arteether vs quinine were restricted to African children with cerebral malaria.

All the studies were at low risk of deviation from interventions, had nearly all outcome data available, and low risk of selective reporting as shown in Fig A in S1 File. About 60% (n = 19) of the RCTs had adequate randomisation and 70% (n = 23) had concerns with measurement of outcomes. If outcome assessors or microscopists were not blinded, the study was considered with concerns as assessing parasite count and coma recovery involves some subjectivity.

The NMA among children included all the treatments whilst the adult's did not involve arteether as seen in Fig 2. Summary RRs or MDs with 95%CIs for all possible comparisons are shown in Table 3 for children (above diagonal) and adults (below diagonal). Fig 3 (children) and Fig 4 (adults) show ranking of treatments by probability of being the best against quinine.

For mortality, data for children involved 20 RCTs and 7534 participants. Among children, artesunate reduced mortality compared to quinine (RR, 0·76; 95%CI [0·65 to 0·89]), artemether (RR, 0·81; 95%CI [0·62 to 1.07]) and artemisinin (RR, 0·83; 95%CI [0·26 to 2.67]), there was no evidence of heterogeneity or inconsistency. For adults there were 13 RCTs, pooling 3399 participants. Both artemether (RR, 0·60; 95%CI [0·42 to 0·85]) and artesunate (RR, 0·55; 95%CI [0·40 to 0·75]) significantly reduced mortality compared to quinine. Additionally, artesunate was better than artemether (RR, 0·91; 95%CI [0·61 to 1.37]) and artemisinin (RR, 0·61; 95%CI [0·32 to 1.17]). There was also no evidence of heterogeneity or inconsistency.

Data for coma recovery time involved 515 children and 671 adults. Artemether (MD; hours, -11·98; 95%CI [-22·21 to -1·75]) showed shorter coma recovery time than arteether in

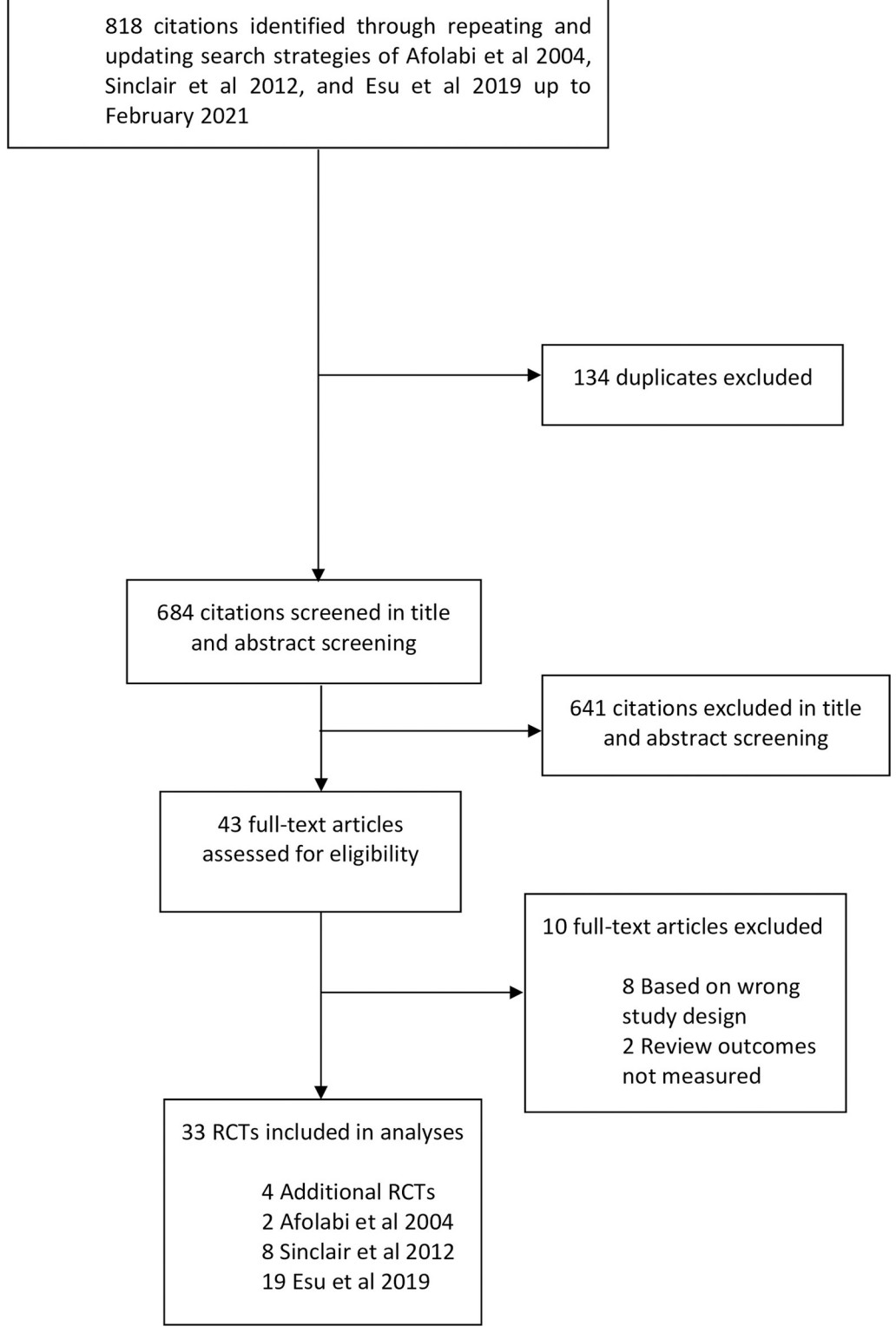

**Fig 1. Study flow diagram.**

**Table 1. Study characteristics.**

| Author Year | Trial Date | N | % Female | Country | Cerebral Malaria Only | Follow up(days) | Drugs | Route of Administration | Age Group | Age[a] (years) |
|---|---|---|---|---|---|---|---|---|---|---|
| Anh et al 1989 [44] | Feb to Dec 1989 | 41 | 24 | Vietnam | Yes | unclear | ASU | IM | Adults | 30.26 ± 13.75 |
| | | | | | | | QN | IV | | 38.68 ± 13.05 |
| Win et al 1992 [42] | Feb 1989 to Aug 1991 | 141 | N/R | Myanmar | Yes | unclear | ASU | IV | Adults | 23.67 ± 4.8 |
| | | | | | | | AME | IM | | 22.98 ± 2.62 |
| | | | | | | | QN | IV | | 22.23 ± 3.87 |
| Karbwang et al 1992 [45] | May to Dec 1991 | 26 | 4.0 | Thailand | No | 7 | AME | IM | Adults | 30.4 ± 10 |
| | | | | | | | QN | IV | | 31.7 ± 10.4 |
| Hien et al 1992 [24] | 1989 to 1990 | 79 | 11 | Vietnam | Yes | unclear | ASU | IV | Adults | 29 (16–50)[R] |
| | | | | | | | AMI | Rectal | | 30 (15–50)[R] |
| | | | | | | | QN | IV | | 28 (19–52)[R] |
| Walker et al 1993 [46] | N/R | 54 | 44 | Nigeria | Yes | >28 | AME | IM | Children | **N/R** |
| | | | | | | | QN | IV | | **N/R** |
| Anh et al 1995 [47] | July 1992 to May 1995 | 190 | 18 | Vietnam | Yes | unclear | ASU | IM | Adults | 32.8 (17–62)[R] |
| | | | | | | | QN | IV | | 29.1 (16–63)[R] |
| Karbwang et al 1995 [48] | 1992 to 1994 | 102 | 10 | Thailand | No | unclear | AME | IM | Adults | 25[M] (15–55)[R] |
| | | | | | | | QN | IV | | 28[M] (15–54)[R] |
| Hien et al 1996 [49] | May 1991 to June 1996 | 561 | 24 | Vietnam | No | unclear | AME | IM | Adults | 30[M] (15–79)[R] |
| | | | | | | | QN | IM | | 30[M] (15–78)[R] |
| van Hensbroek et al 1996 [50] | 1992 to 1994 | 576 | 49 | Gambia | Yes | 28 | AME | IM | Children | 4 ± 1.8 |
| | | | | | | 28 | QN | IM | | 3.8 ± 1.8 |
| Murphy et al 1996 [51] | N/R | 160 | 50 | Kenya | Yes | unclear | AME | IM | Children | 2.1[M] (1.2–9)[R] |
| | | | | | | | QN | IV | | 2.5[M] (1.2–12)[R] |
| Vihn et al 1997 [25] | 1992 to 1994 | 180 | 32 | Vietnam | No | unclear | ASU | IV | Adults | 30[M] (15–60)[R] |
| | | | | | | | ASU | IM | | 24[M] (15–66)[R] |
| | | | | | | | AMI | Rectal | | 28[M] (16–62)[R] |
| | | | | | | | AME | IM | | 28[M] (16–65)[R] |
| Phuong et al 1997 [26] | Aug 1992 to Mar 1995 | 109 | 50 | Vietnam | No | 28 | ASU | IM | Children | 6[M] (0.5–14)[R] |
| | | | | | | | AMI | Rectal | | 7[M] (0.7–14)[R] |

*(Continued)*

**Table 1.** (Continued)

| Author Year | Trial Date | N | % Female | Country | Cerebral Malaria Only | Follow up(days) | Drugs | Route of Administration | Age Group | Age[a] (years) |
|---|---|---|---|---|---|---|---|---|---|---|
| | | | | | | | QN | IV | | 5 [M] (0·3–13) [R] |
| Seaton et al 1998 [52] | Jun 1992 to May 1995 | 33 | N/R | Papua New Guinea | No | 28 | AME | IM | Adults | N/R |
| | | | | | | | QN | IV | | |
| Taylor et al 1998 [53] | Jan 1992 to Jun 1994 | 183 | 45 | Malawi | Yes | >28 | AME | IM | Children | 2·9 ± 2 |
| | | | | | | | QN | IV | | 3·2 ± 2 |
| Ojuawo et al 1998 [54] | unclear | 37 | N/R | Nigeria | Yes | unclear | AME | IM | Children | N/R |
| | | | | | | | QN | IV | | |
| Olumese et al 1999 [55] | not stated | 103 | 47 | Nigeria | No | 28 | AME | IM | Children | 3·1 (N/R) |
| | | | | | | | QN | IV | | 3.2 (N/R) |
| Thuma et al 2000 [56] | Jan 1996 to May 1997 | 95 | 47 | Zambia | Yes | 28 | ATE | IM | Children | 3·9 ± 2·3 |
| | | | | | | | QN | IV | | 3·3 ±1·8 |
| Moyou-Somo et al 2001 [57] | Nov 1995 to Dec 1997 | 102 | 42 | Cameroon | Yes | 28 | ATE | IM | Children | 3·4 (N/R) |
| | | | | | | | QN | IV | | 3·3 (N/R) |
| Adam et al 2002 [58] | Nov 2001 to Jan 2002 | 41 | 49 | Sudan | No | >28 | AME | IM | Children | 4·1(2·5) |
| | | | | | | | QN | IV | | 3·59(3·2) |
| Satti et al 2002 [59] | May 1995 to Jun 1996 | 77 | N/R | Sudan | Yes | 28 | AME | IM | Children | N/R |
| | | | | | | | QN | IV | | |
| Newton et al 2003 [60] | May to Jul 1994 and 1995 to 2001 | 113 | 43 | Thailand | No | unclear | ASU | IV | Adults | 25 [M] (15–66) [R] |
| | | | | | | | QN | IV | | 25 [M] (15–59) [R] |
| Huda et al 2003 [61] | Apr 2000 to Jul 2001 | 46 | 48 | India | No | 28 | AME | IM | Children | 6·6 ± 3·5 |
| | | | | | | | QN | IV | | 5·8 ± 2·4 |
| Mohanty et al 2004 [40] | Jan 2000 to Jan 2002 | 80 | 40 | India | No | 28 | AME | IM | Children | 8·1 ± 3·23 |
| | | | | | | | QN | IV | | 7·31 ± 3·47 |
| Minta et al 2005 [62] | Jun 1993 to Feb 1994 and Jun 1994 to Dec 1994 | 67 | N/R | Mali | No | unclear | AME | IM | Children | 7·3 ± 3·9 |
| | | | | | | | QN | IV | | 6·3 ± 4·0 |
| Dondorp et al 2005 [63] | Jun 2003 to May 2005 | 1761 | 30 | Bangladesh, Myanmar (Burma), India, and Indonesia. | No | >28 | ASU | IV | Both | 27·9 (N/R) |
| | | | | | | | QN | IV | | 27·9 (N/R) |
| Haroon et al 2005 [43] | July 2000 to Aug 2002 | 35 | 14 | India | No | unclear | AME | IM | Adults | 32 [M] (18–47·5) [I] |
| | | | | | | | QN | IV | | 31 [M] (18–47·5) [I] |
| Aguwa et al 2010 [64] | Jul to Oct 2007 | 90 | 58 | Nigeria | N0 | 14 | AME | IM | Children | 3·2 ± 1·7 |
| | | | | | | | QN | Parenteral | | 3·8 ± 1·3 |

*(Continued)*

**Table 1.** (Continued)

| Author Year | Trial Date | N | % Female | Country | Cerebral Malaria Only | Follow up(days) | Drugs | Route of Administration | Age Group | Age[a] (years) |
|---|---|---|---|---|---|---|---|---|---|---|
| Phu et al 2010 [65] | May 1996 to Jun 2003 | 370 | 26 | Vietnam | No | unclear | AME | IM | Adults | 32·5 [M] (15–77) [R] |
| | | | | | | | ASU | IM | | 32 [M] (15–74) [R] |
| Dondorp et al 2010 [5] | Oct 2005 to Jul 2010 | 5425 | 48 | Mozambique, The Gambia, Ghana, Kenya, Tanzania, Nigeria, Uganda, Rwanda, and Democratic Republic of the Congo | No | >28 | ASU | Parenteral | Children | 2.8 [M] (1·6–4·2) [I] |
| | | | | | | | QN | Parenteral | | 2·9 [M] (1·7–4·3) [I] |
| Eltahir et al 2010 [66] | Aug to Sep 2010 | 66 | 44 | Sudan | No | unclear | ASU | IV | Children | 4·4 ± 2·6 |
| | | | | | | | QN | IV | | 4·6 ± 3·4 |
| Osunuga et al 2011 [67–69] | N/R | 32 | 38 | Nigeria | No | 14 | AME | IM | Children | 6 ± 3·7 |
| | | | | | | | QN | IV | | 8·2 ± 3·4 |
| Abdallah et al 2014 [41] | Oct 2012 to Dec 2012 | 94 | 43 | Sudan | No | unclear | ASU | IV | Both | 23·5 ± 20·2 |
| | | | | | | | QN | IM | | 21·5 ± 17·6 |
| Bobossi-Serengbe et al 2015 [70] | Jun to Aug 2010 | 212 | 55 | Central African Republic | No | unclear | AME | IM | Children | 2·4 (N/R) |
| | | | | | | | QN | IV | | 2·5 (N/R) |

AMI, Artemisinin; ATE, Arteether; AME, Artemether; ASU, Artesunate; QN, Quinine; NR, Not Reported.
[a]Age was reported in mean ± SD unless superscript is provided, [R] Range, [I] Interquartile range, [M] Median.

children. Artemether had shorter coma recovery time than quinine in both children (MD; hours, -5·29; 95%CI -7·94 to -2·64]) and adults (MD; hours, -2·82; 95%CI [-18·89 to -13·25]). Data was not available for artemisinin and artesunate for the children and adult analyses respectively. There was evidence of heterogeneity ($I^2$ = 64%, p-value = 0.1) and inconsistency ($I^2$ = 52%, p-value = 0.15) in the adult analyses. To further explain the heterogeneity, and inconsistency we used net splitting method in Fig B in S1 File which showed an overlap between the direct and indirect evidence for all designs. However, a net heat plot was not available due to small number of designs.

The analyses of parasite clearance time involved 11 RCTs (718 participants) for children and four RCTs (656 participants) for adults. Data was not available for artemisinin in the children analysis and arteether in the adult analysis. Artemether showed a shorter parasite clearance time than quinine in both children (MD; hours, -7·43; 95%CI -11·40 to -3·46]) and adults (MD; hours, -14·45; 95%CI -28·60 to -0·31]). Artesunate also reduced parasite clearance time compared to quinine in children (MD; hours, -1·10; 95%CI -9·50 to 7·30]) and adults (MD; hours, -9·42; 95%CI -20·60 to 1·25]). Artemisinin was better than quinine in clearing parasites (MD; hours, -5·51; 95%CI -24·13 to 13·11]) in adults. Both artemether and artesunate also recorded a better parasite clearance time than arteether in children.

There was moderate heterogeneity ($I^2$ = 58%, p = 0·04) present in the children analyses and high inconsistency ($I^2$ = 84%, p< 0·01) in the adult analysis.

**Table 2. Summary of characteristics for all included RCTs in age group and type of severe malaria.**

| | Overall | | Age group* Children | | Adult | | Type of severe malaria Cerebral malaria only | | Non-Specified | |
|---|---|---|---|---|---|---|---|---|---|---|
| | n | % | n | % | n | % | n | % | n | % |
| **Total studies** | 33 | 100 | 19 | 58 | 12 | 36 | 12 | 36 | 21 | 64 |
| Publication Year | | | | | | | | | | |
| 1989–1999 | 16 | 49 | 7 | 37 | 9 | 75 | 9 | 75 | 7 | 33 |
| 2000–2009 | 10 | 30 | 7 | 37 | 2 | 17 | 3 | 25 | 7 | 33 |
| 2010–2019 | 7 | 21 | 5 | 26 | 1 | 8 | 0 | 0·0 | 7 | 33 |
| Participants | | | | | | | | | | |
| <100 | 17 | 51 | 11 | 58 | 5 | 42 | 6 | 50 | 11 | 52 |
| ≥ 100 | 16 | 49 | 8 | 42 | 7 | 58 | 6 | 50 | 10 | 48 |
| Age group | | | | | | | | | | |
| children | 19 | 58 | 19 | 100 | 0 | 0·0 | 8 | 67 | 11 | 52 |
| adults | 12 | 36 | 0 | 0·0 | 12 | 100 | 4 | 33 | 8 | 38 |
| both | 2 | 6 | 0 | 0·0 | 0 | 0·0 | 0 | 0·0 | 2 | 10 |
| Type of Severe Malaria | | | | | | | | | | |
| cerebral malaria only | 12 | 36 | 8 | 42 | 4 | 33 | 12 | 100 | 0 | 0·0 |
| non-specified | 21 | 64 | 11 | 58 | 8 | 67 | 0 | 0·0 | 21 | 100 |
| Study Continent | | | | | | | | | | |
| Africa | 17 | 51 | 16 | 84 | 0 | 0·0 | 8 | 67 | 9 | 43 |
| Asia | 15 | 46 | 3 | 16 | 11 | 92 | 4 | 33 | 11 | 52 |
| **South Pacific | 1 | 3 | 0 | 0·0 | 1 | 8 | 0 | 0·0 | 1 | 5 |
| Follow up (days) | | | | | | | | | | |
| < 28 | 4 | 12 | 3 | 16 | 1 | 8 | 1 | 8 | 3 | 14 |
| ≥ 28 | 15 | 46 | 13 | 68 | 1 | 8 | 5 | 42 | 8 | 38 |
| unclear | 14 | 42 | 3 | 16 | 10 | 83 | 6 | 50 | 10 | 48 |
| ***Treatments | | | | | | | | | | |
| Artemether | 22 | 67 | 14 | 74 | 8 | 67 | 7 | 58 | 15 | 71 |
| Artemisinin | 3 | 9 | 1 | 5 | 2 | 17 | 1 | 8 | 2 | 10 |
| Arteether | 2 | 6 | 2 | 11 | 0 | 0·0 | 2 | 16 | 0 | 0·0 |
| Artesunate | 12 | 36 | 3 | 16 | 7 | 58 | 4 | 33 | 8 | 38 |
| Quinine | 31 | 94 | 19 | 100 | 10 | 83 | 12 | 100 | 19 | 90 |

This table displays column percentages except for first row.

*Does not add up to 100% since two studies included both age groups and were omitted.

** Study conducted in Papua New Guinea.

***Arm level data that do not add up to the total number of studies.

Further examination of inconsistency in the adult analyses showed overlaps between direct and indirect evidence, however, treatment effects were mostly in different directions as seen in Fig C I in S1 File. The following designs were the largest contributors to the inconsistency: quinine vs artesunate, quinine vs artemether, and the quinine vs artemether vs artesunate comparisons in the net heat plot (Fig C II in S1 File).

Except for the comparison of artemether vs quinine in children (MD; hours, -7·92; 95%CI [-13·21 to -2·63]), there were no significant differences in fever clearance time between the drugs in either children (12 RCTs, 749 participants), or adults (4 RCTs, 656 participants). Artesunate (MD; hours, -8·18; 95%CI [-32.20 to 15.84]), artemether (MD; hours, -14.98; 95%CI

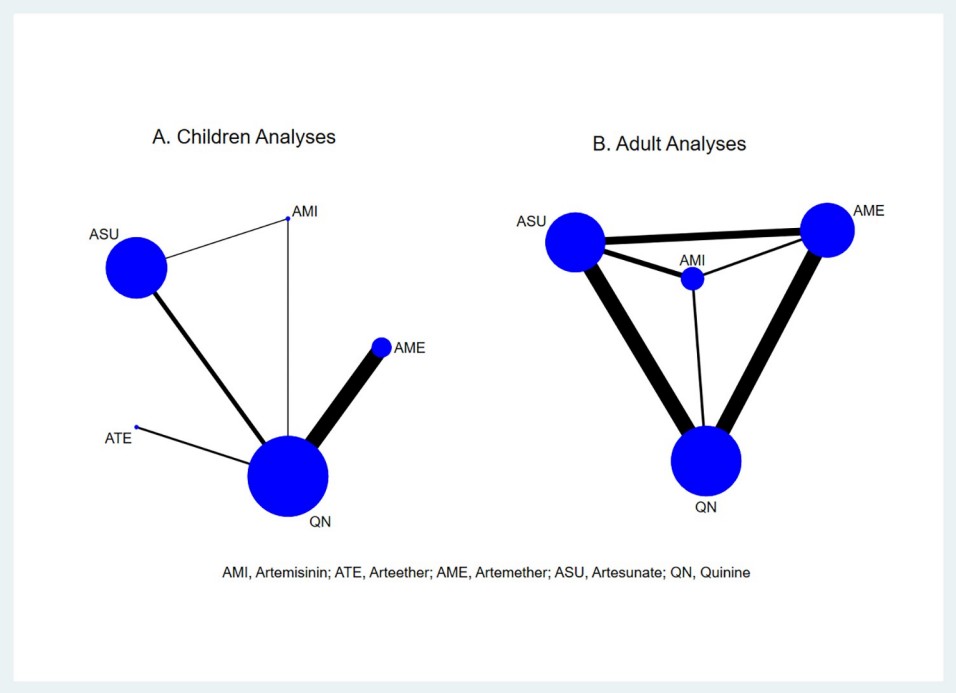

**Fig 2.** Network graph of treatment comparisons for mortality among A) children and B) adults. The blue nodes are proportional to the number of participants allocated to that drug. The thickness of the black edges is proportional to the number of studies comparing the drugs on each side of the edge.

[-32.31 to 2·68]) and quinine (MD; hours, -6.18.49; 95%CI [-22.47 to 10·11]), reduces fever clearance time compared to arteether among children. There was substantial evidence of both heterogeneity in the children analysis ($I^2$ = 81%, p<0·01) and inconsistency in the adult analysis ($I^2$ = 74%, p<0·01) for this outcome. A further look at the inconsistency in the adult analysis showed that the 95%CI of both direct and indirect effect sizes mostly overlapped while the net heat plots showed hotspots involving quinine vs artemether, quinine vs artesunate, and quinine vs artemether vs artesunate designs (Figs D I and II in S1 File).

The neurological sequelae events mentioned in some of the RCTs were blindness, deafness, facial paresis, paresis, aphasia, ataxia, mental retardation, loss of neurological milestones, myasthenia gravis-like syndrome, and hallucinations [5,44–48]. We have converted ORs in Figs E and G in S1 File to RRs for easy interpretation but the overall direction on the forest plots remain the same. Traditional meta-analysis pooling 10 RCTs showed that artemether (OR, 0·87; 95%CI [0·55 to 1·37], RR, 0·87; 95%CI [0·56 to 1·34]) may reduce neurological sequela events compared to quinine with mild heterogeneity ($I^2$ = 28%, p = 0·22). Results can be seen in Fig E in S1 File.

Eighteen studies reported adverse events. Hypoglycemia was the most reported adverse effect with 11 RCTs (8953 participants). The network forest plot combining events in both age groups displayed in Fig F in S1 File, shows that artemether (RR, 0·53; 95%CI [0·40 to 0·70]), artesunate (RR, 0·53; 95%CI [0·40 to 0·70]), and artemisinin (RR, 0·30; 95%CI [0·09 to 0·96]) reduced the occurrence of hypoglycaemia during treatment compared to quinine. There was no evidence of heterogeneity or inconsistency. There was no data available for arteether.

Four individual studies reported electrocardiogram abnormalities; all comparing artemether to quinine. A traditional meta-analysis conducted as shown in Fig G in S1 File, found that artemether (OR, 1·72; 95%CI [0·96 to 3·05], RR, 1·65; 95%CI [0·96 to 2·75]) may increase

**Table 3. League table of NMA results with measures of variability [I$^2$ Statistic, p-value] for children and adult analyses.**

**Mortality; RR (95% CI)**

| | | | Tot;0·0%, p = 0·90 | Het;0·0%, p = 0·90 |
|---|---|---|---|---|
| n = 20 and N = 7534 | | | | Inc;0·0%, p = 0·37 |
| ASU | 0·81 (0·62; 1·07) | 0·83 (0·26; 2·67) | 1·00 (0·56; 1·79) | 0·76 (0·65; 0·89) |
| 0·91 (0·61; 1·37) | AME | 1·03 (0·32; 3·34) | 1·24 (0·68; 2·25) | 0·93 (0·75; 1·17) |
| 0·61 (0·32; 1·17) | 0·67 (0·33; 1·34) | AMI | 1·20 (0.33; 4.36) | 0·91 (0·29; 2·90) |
| - | - | - | ATE | 0·76 (0·43; 1·32) |
| 0·55 (0·40; 0·75) | 0·60 (0·42; 0·85) | 0·90 (0·46; 1·75) | - | QN |
| Tot;24%, p = 0·20 | Het;24%, p = 0·25 | | | |
| Inc;23%, p = 0·25 | | | n = 13 and N = 3399 | |

**Coma Recovery Time MD, hours (95% CI)**

| | | | Tot; 0%, p = 0·14 | Het;0%, p = 0·14 |
|---|---|---|---|---|
| n = 10 and N = 515 | | | | Inc; 0%, p = NA |
| ASU | - | - | - | - |
| -0·58 (-16·63; 15·47) | AME | - | -11·98 (-22·21; -1·75) | -5·29 (-7·94; -2·64) |
| - | - | AMI | - | - |
| - | - | - | ATE | 6·69 (-3·19; 16·57) |
| -3·40 (-15·53; 8·72) | -2·82 (-18·89; -13·25) | - | - | QN |
| Tot;59%, p = 0·09 | Het;64%, p = 0·1 | | | |
| Inc;52%, p = 0·15 | | | n = 3 and N = 671 | |

**Parasite Clearance Time; MD, hours (95% CI)**

| | | | Tot;58%, p = 0·04 | Het;58%, p = 0·04 |
|---|---|---|---|---|
| n = 11 and N = 718 | | | | Inc;0%, p = NA |
| ASU | 6·33 (-2·96; 15·62) | - | -1·56 (-14·26; 11·15) | -1·10 (-9·50; 7·30) |
| 5·03 (-10·47; 20·53) | AME | - | -7·89 (-18·21; 2·44) | -7·43 (-11·40; -3·46) |
| -3·91 (-22·10; 14·28) | -8·94 (-31·41; 13·53) | AMI | - | - |
| - | - | - | ATE | 0·46 (-9·07; 9·99) |
| -9·42 (-20·10; 1·25) | -14·45 (-28·60; -0·31) | -5·51 (-24·13; 13·11) | - | QN |
| Tot;84%, p<0·01 | Het;0%, p = NA | | | |
| Inc;84%, p<0·01 | | | n = 4 and N = 656 | |

**Fever Clearance Time MD, hours (95% CI)**

| | | | Tot;81%, p<0·01 | Het;81%, p<0·01 |
|---|---|---|---|---|
| n = 12 and N = 749 | | | | Inc;0%, p = NA |
| ASU | 6·64 (-12·13; 15·84) | - | -8·18 (-32·20; 15·84) | -2·00 (-19·65; 15·65) |
| -2·07 (-27·31; 23·16) | AME | - | -14·98 (-32·31; 2·68) | -7·92 (-13·21; -2·63) |
| -31·92 (-68·23; 4·38) | -29·85 (-72·73; 13·04) | AMI | - | - |
| - | - | - | ATE | 6·18 (-10·11; 22·47) |
| -12·54 (-31·94; 6·86) | -10·46 (-33·06; 12·14) | 19·39 (-19·62; 58·39) | - | QN |
| Tot;74%, p<0·01 | Het;0%, p = NA | | | |
| Inc;74%, p<0·01 | | | n = 4 and N = 656 | |

RR, Risk Ratio; MD, Mean Difference; AMI, Artemisinin; ATE, Arteether (only in children); AME, Artemether; ASU, Artesunate; QN, Quinine; Tot, Total Variability;

Het, Heterogeneity; Inc, Inconsistency; p, p-values; n, number of RCTs; N, number of participants.

The upper diagonal provides summary of network estimates (both direct and indirect results) among children.

It is read as the treatment to the left of estimates versus treatment beneath estimates. With measures of variability in upper right corner.

The lower diagonal provides summary of network estimates (both direct and indirect results) among adults.

It is read as the treatment to the left of estimates *versus* the treatment to the right.

With measures of variability in lower left corner.

The DerSimonian and Laird method was used to estimate the between-study variance, and the Jacksons method to estimate its confidence intervals [31,35].

P-values were two-sided with a significance level of 0.05.

Cochran Q method was used to assess the total variability and then decomposed into heterogeneity and inconsistency [36].

the number of electrocardiogram abnormalities compared to quinine, and this was associated with moderate heterogeneity (I$^2$ = 69%, p = 0·02).

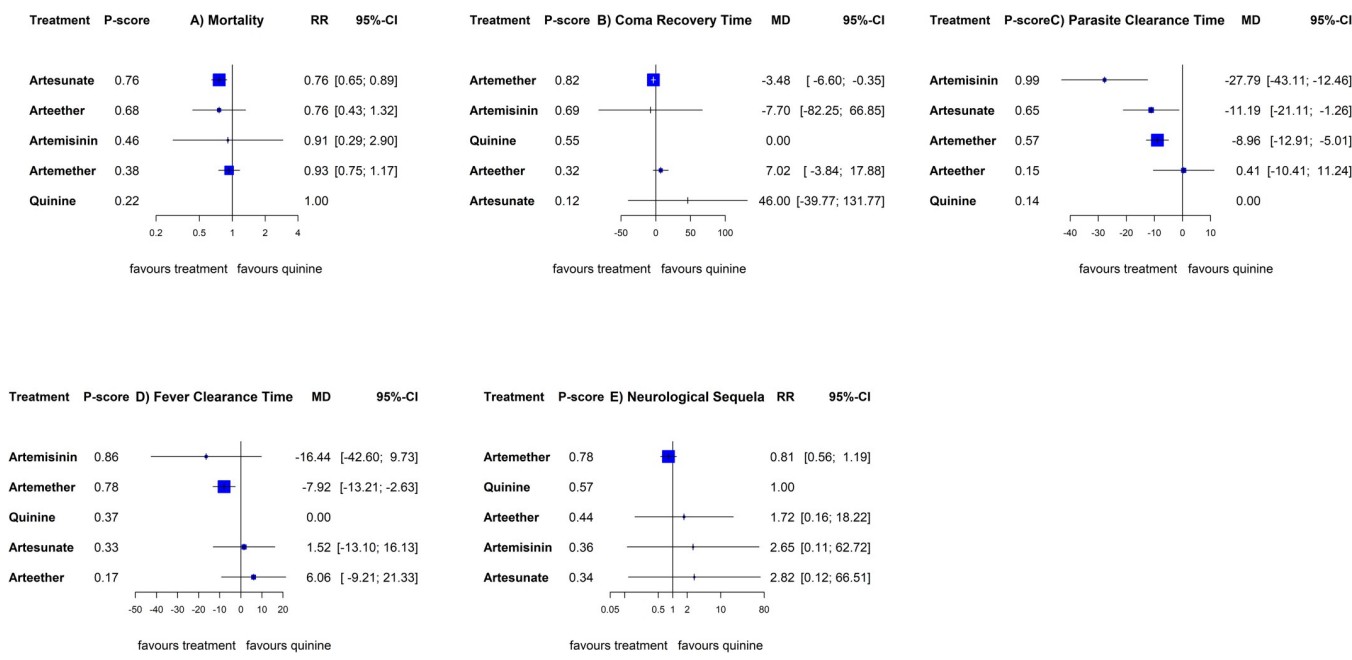

**Fig 3.** Forest plots comparing artemisinin derivatives against quinine (reference) for A) mortality, B) coma recovery time, C) parasite clearance time, D) fever clearance time, and E) neurological sequelae in children. Treatments were ranked by probability of being the best for that given outcome. Abbreviations: RR: Risk Ratio; MD: Mean Difference; CI: Confidence Interval.

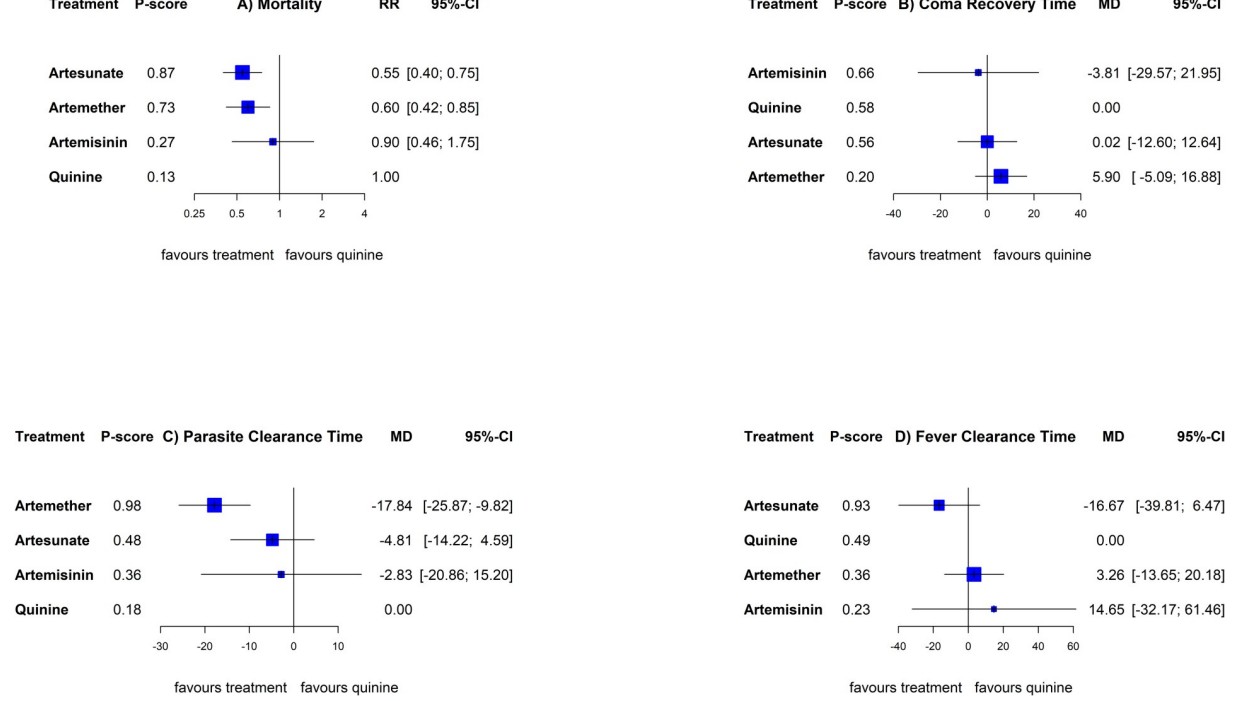

**Fig 4.** Forest plots comparing artemisinin derivatives against quinine (reference) for A) mortality, B) coma recovery time, C) parasite clearance time, and D) fever clearance time in adults. Treatments were ranked by probability of being the best for that given outcome. Abbreviations: RR: Risk Ratio; MD: Mean Difference; CI: Confidence Interval.

Subgroup analyses were conducted with children, and adults together. Table D in S1 File displays the full NMA results of all subgroup analyses. In 14 RCTs (4321 participants) reporting cerebral malaria only, artesunate (RR, 0·72; 95%CI [0·55 to 0·94]) significantly reduced mortality compared to quinine. The 16 RCTs carried out in Asia showed that artesunate (RR, 0·61; 95%CI [0·49 to 0·76]), and artemether (RR, 0·66; 95%CI [0·50 to 0·86]) both reduced mortality. Whilst 17 RCTs conducted in Africa showed only artesunate (RR, 0·80; 95%CI [0·67 to 0·95]) significantly lowered mortality compared to quinine. Only six and four RCTs reported acute and persistent neurological sequela events respectively. Artemether reduced occurrence of acute events compared to artesunate, and quinine (RR, 0·74; 95%CI [0·57 to 0·97] and RR, 0·55; 95%CI [0·36 to 0·82] respectively. The other drugs apart from artemether increased the occurrence of events compared to quinine. There were no results available for rectal artemisinin.

For sensitivity analyses, there were broadly similar findings for the analyses involving the addition of studies in which conversions were made and the combination of age groups, as compared to the main analyses. The full results are shown in Figs H, I and Table E in S1 File. Publication bias as assessed by Egger's test (p = 0.03) was significant, and there was slight asymmetry in the comparison adjusted funnel plot (Fig 5). Smaller studies were more likely to produce a beneficial effect in favour of all the artemisinin drugs.

The quality of evidence generated for artesunate vs quinine was moderate for both children and adults. All the other comparisons among children, and adults were mostly of very low ratings. The ratings of evidence are presented in Data B and Table F in S1 File.

## Discussion

This network meta-analysis used state-of-the-art methods to combine data from 33 RCTs (10977 participants) to provide evidence for the efficacy of artemisinin drugs for treating severe malaria, including cerebral malaria, across clinically relevant outcomes. Artesunate significantly reduced the risk of mortality in children by 24% (95%CI 11% to 35%) compared to quinine, whereas for adults, both artesunate and artemether showed a reduction of 45% (95% CI 25% to 60%) and 40% (95%CI 15% to 58%) respectively. Artesunate also reduced mortality in cerebral malaria by 28% (95%CI 6% to 45%) compared to quinine. None of the artemisinin drugs was consistently superior across all outcomes for both adults and children.

This review extends findings from other systematic reviews with traditional meta-analysis showing that parenteral artesunate is better than quinine in reducing mortality and clearing parasites in both age groups [16,63]. Artesunate was the only artemisinin derivative that consistently reduced mortality compared to quinine for both age groups in Asia, and Africa including cerebral malaria. An earlier systematic review could not establish the efficacy of the artemisinin derivatives in treating cerebral malaria. [17] This NMA has provided moderate quality evidence for the efficacy of artesunate in the subgroup analysis, with two additional large RCTs [5,63]. Artesunate has drawn much attention because of its superior pharmacokinetics; it is water soluble, and is rapidly and reliably absorbed after administration, and so, has benefitted from many large studies [5,63]. However, we have observed in this NMA that its clinical superiority is not clearly established compared to the other artemisinin derivatives.

Just like other reviews, we have also observed that IM artemether was better than quinine in shortening coma recovery and clearing parasites in both age groups. It also reduced mortality among children, but may not be effective in reducing mortality in cerebral malaria [11,14]. Possibly because it has an erratic absorption especially in the severely ill [8].

Among adults, rectal artemisinin showed shorter parasite clearance time compared to quinine. There was insufficient evidence to support this drug being superior in the other

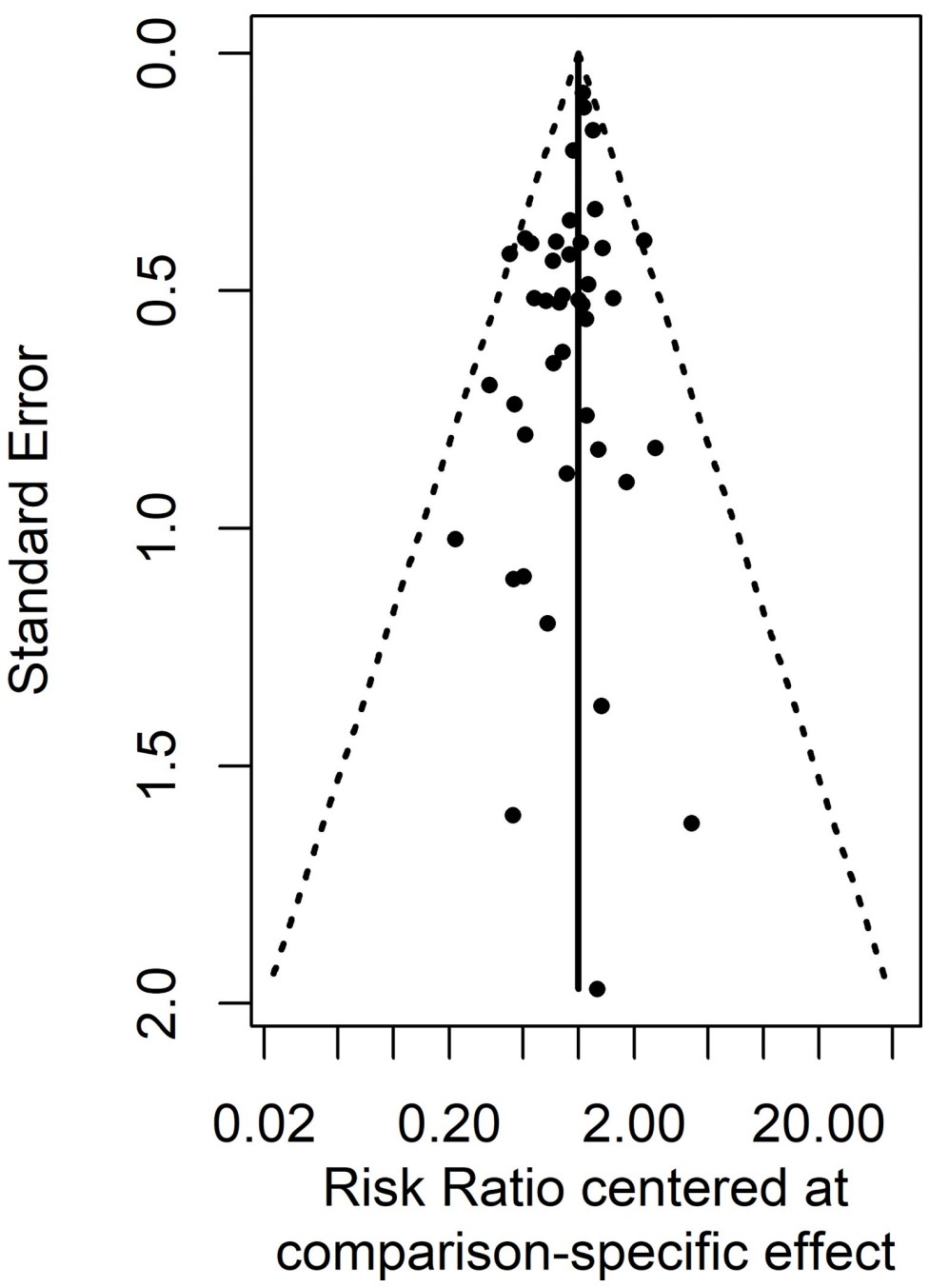

**Fig 5. Comparison adjusted funnel plot.**

outcomes. Rectal artemisinin has shown potential and can provide easier to administer formulations during community-based emergency treatments especially, in under-resourced settings [71]. There was limited evidence to suggest IM arteether was better than quinine for any of the outcomes investigated [15]. IM arteether has been registered and is used in the Netherlands for children only and in India for both children, and adults for treating severe *P.falciparum* malaria. Arteether was produced as a cheaper alternative for developing countries [72,73].

The artemisinin derivatives were not associated with more hypoglycaemia events, or electrocardiogram abnormalities. All artemisinin drugs except artemether may increase neurological sequela events in the first three to seven days of treatment, compared to quinine. Most of the neurological sequela events resolved during follow up, which is similar to findings of other studies [5,42,53]. However, we observed that mortality is a competing event with these adverse outcomes.

We have combined direct, and indirect evidence to compare the artemisinin derivatives and quinine in treating severe malaria. This review has produced a robust estimate of effect sizes for all comparisons between the artemisinin derivatives in treating severe malaria and examined the impact of treatment across a range of outcomes, in the absence of RCTs. We carried out thorough risk of bias assessments, used the latest tools to assess quality of evidence [29,39] and used sensitivity analyses to explore the impact of approximations made from medians and ranges/ interquartile ranges [28]. We conducted separate analyses for children and adults to reduce clinical heterogeneity and increase clinical relevance of the results.

The evidence produced was of very low to moderate quality. Several limitations contributed to this. Apart from mortality, reporting other outcomes was not consistent for all RCTs, which contributed to loss of power to adequately analyse secondary outcomes. Mortality may be considered as a competing event with some of the secondary outcomes but we were unable to explore this using more appropriate methods because individual participant data was not available. Majority of the RCTs used in this review were judged as moderate risk because it was not clear whether assessors for secondary outcomes were blinded. The interventions were not evenly distributed among the study continents and age groups, however, this did not manifest statistically in the primary outcome. We detected mild to important statistical inconsistency in secondary outcomes using the recommended approach of combining methods [74]. A detailed discussion of inconsistency can be seen in Data C S1 File. There were few trials for arteether and rectal artemisinin with small sample sizes, which meant that the analyses for these drugs were available for few comparisons and underpowered to detect any clinically significant effect sizes. The ranking in the frequentist framework, using the p-scores is largely based on point estimates and less on precision so, interpretation should also consider the 95%CI. We chose not to use the Hasse diagram to combine rankings as prespecified in our protocol because it is not possible to give weight to most important outcomes and it compounds the limitations of the p-scores [38,75].

We detected an unequal distribution of RCTs in age groups among Asian and African populations; sixteen out of 17 RCTs from Africa were conducted among children and 11 out of 12 RCTs conducted among adults were in Asia. The earliest systematic reviews found that artemether was more effective among Asian adults than in African children [13,18]. This has partly motivated subsequent RCTs along these subgroups. Even though efforts were made to obtain unpublished data, we had concerns with reporting bias. These are all sources of biases that are inherent in all severe malaria systematic reviews comparing the artemisinin drugs.

The artemisinin derivatives are already used worldwide, we have provided prescribers a broader understanding of how the artemisinin derivatives compare in improving mortality, coma recovery, parasite count, fever, neurological sequela, and hypoglycemia during severe malaria treatment. Artesunate and artemether are better than quinine for severe malaria treatment but their life saving benefits are not clear compared to the other artemisinin drugs. IM arteether and rectal artemisinin are not used widely and have not benefitted from any large trials, yet they have so much potential.

Artemisinin resistance becomes more likely when optimal treatment and doses are not used. A return to widespread quinine use with artemisinin treatment failure would be a major threat to malaria elimination and could cause 230,000 additional severe malaria cases and

116,000 excess deaths every year with an estimated cost of 385 million US$ [9]. There is a pressing need for effectiveness and pragmatic trials to inform the proper use of the artemisinin derivatives in severe malaria treatment. Our findings are useful in drug development, exploring new artemisinin-based combinations, improving future phase three RCTs, informing treatment guidelines, and managing artemisinin drug resistance.

## Supporting information

**S1 Checklist. PRISMA checklist.**
(PDF)

**S1 File. All supplementary data, tables and figures.**
(PDF)

## Acknowledgments

We thank Dr. Sarah Donegan and Dr. Ekpereonne Esu for sharing unpublished studies from past Cochrane systematic reviews.

## Author Contributions

**Conceptualization:** Nicholas Nyaaba, José M. Ordóñez-Mena, Jennifer Hirst.

**Data curation:** Nicholas Nyaaba, Nana Efua Andoh, Gordon Amoh, Dominic Selorm Yao Amuzu, Mary Ansong, José M. Ordóñez-Mena, Jennifer Hirst.

**Formal analysis:** Nicholas Nyaaba, José M. Ordóñez-Mena, Jennifer Hirst.

**Methodology:** Nicholas Nyaaba, José M. Ordóñez-Mena, Jennifer Hirst.

**Resources:** Nicholas Nyaaba, José M. Ordóñez-Mena, Jennifer Hirst.

**Supervision:** Nicholas Nyaaba, José M. Ordóñez-Mena, Jennifer Hirst.

**Validation:** Nicholas Nyaaba, Nana Efua Andoh, Gordon Amoh, Dominic Selorm Yao Amuzu, Mary Ansong, José M. Ordóñez-Mena, Jennifer Hirst.

**Writing – original draft:** Nicholas Nyaaba, José M. Ordóñez-Mena, Jennifer Hirst.

**Writing – review & editing:** Nicholas Nyaaba, Nana Efua Andoh, Gordon Amoh, Dominic Selorm Yao Amuzu, Mary Ansong, José M. Ordóñez-Mena, Jennifer Hirst.

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
