## [Decision Letter · Decision Letter 0]

12 Nov 2021

PONE-D-21-29782Comparative efficacy and safety of the Artemisinin Derivatives compared to Quinine for treating Severe Malaria in Children and Adults: a Systematic Update of Literature and Network Meta-analysisPLOS ONE

Dear Dr. Nyaaba,

Thank you for submitting your manuscript to PLOS ONE. After careful consideration, we feel that it has merit but does not fully meet PLOS ONE’s publication criteria as it currently stands. Therefore, we invite you to submit a revised version of the manuscript that addresses the points raised during the review process.

We look forward to receiving your revised manuscript.

Kind regards,

Benedikt Ley, PhD

Academic Editor

PLOS ONE

Journal Requirements:

“Initial part of this review was done as part of Nicholas Nyaaba’s MSc from the University of Oxford which was supported by the Ghana Education Trust Fund (GETFund).”

“The authors received no funding for this work.”

Reviewers' comments:

Reviewer's Responses to Questions

**Comments to the Author**

1. Is the manuscript technically sound, and do the data support the conclusions?

Reviewer #1: Partly

Reviewer #2: Yes

2. Has the statistical analysis been performed appropriately and rigorously? 

Reviewer #1: No

Reviewer #2: Yes

3. Have the authors made all data underlying the findings in their manuscript fully available?

Reviewer #1: Yes

Reviewer #2: Yes

4. Is the manuscript presented in an intelligible fashion and written in standard English?

Reviewer #1: Yes

Reviewer #2: Yes

5. Review Comments to the Author

Reviewer #1: This systematic review and network meta-analysis aimed to assess the efficacy and safety of the artemisinin derivatives and quinine for treating severe P. falciparum malaria in children and adults by updating previous published Cochrane reviews. I have focused my review on the methodological and statistical aspects of this manuscript.

Major comments:

1. The statistical methods section of this review is lacking many details that are either required for a network meta-analysis or are needed to align with the results presented by the authors. I strongly suggest that the authors consult the paper by Chaimani et al 2017, which provides more details than the PRISMA NMA for conducting a network meta-analysis. The authors should also be aware that the PRISMA guidelines have been updated and PRISMA 2020 (Page 2021) should be used in conjunction with PRISMA NMA.

Chamaini et al. Additional considerations are required when preparing a protocol for a systematic review with multiple interventions. Journal of Clinical Epidemiology. 2017: 83; 65-74.

Page et al. The PRISMA 2020 statement: an updated guideline for reporting systematic reviews. BMJ. 2021; 372:n71.

2. The authors have converted medians (range/inter-quartile range) to mean (Standard deviation) and used this in the main analysis. Median and (range/inter-quartile range) are reported when the data do not have a normal distribution. By converting these data to mean (Standard deviation) the authors are assuming a normal distribution. Converting summary statistics for skewed data to summary staitstics used for normally distributed data doesn’t make sense. The authors should produce a meta-analysis of ratios of geometric means when skewed data are reported on the raw scale. Further details of methods used to produce a meta-analysis of skewed data are available in Higgins et al 2008.

Higgins, White and Anzures-Cabrera. Meta-analysis of skewed data: combining results reported on log-transformed or raw scales. Statistics in Medicine. 2008; 27;6072-92.

3. The methods section would benefit from more subsection, including a section that explicitly states that inclusion/exclusion criteria for the study.

4. The eligibility criteria for a network meta-analysis should clearly consider the transitivity assumption. The transitivity assumption is the fundamental assumption underlying the validity of a network meta-analysis. The authors have not considered the transitivity assumption for their network meta-analysis, which is a fundamental error when undertaking a network meta-analysis.

5. The statistical methods section needs to provide an explanation of how the network diagram is created (i.e. what do the size of the nodes and thickness of the edges represent).

6. The authors conduct a meta-analysis of rare events (i.e. analysis of serious adverse events and neurological sequalae) but haven’t considered methods for the meta-analysis of rare events or how they handled studies with no events in one or more arms. Further details are provided in the Cochrane handbook https://training.cochrane.org/handbook/archive/v6.1/chapter-10#section-10-4-4.

7. In figure 1, 2 studies were excluded because the outcomes relevant to this review were not reported. Did the review authors attempt to contact the original study authors to obtain the necessary outcome data?

8. In the results section the authors state: "A traditional meta-analysis conducted as shown in S11, found that artemether…" but details of this analysis are not provided in the methods section.

9. Methods used to estimate the between-study heterogeneity and calculation of the 95% CIs needs to be explicitly stated in the methods section of the review.

10. Subgroup and sensitivity analyses need to be explicitly stated in the statistical methods section.

11. Throughout the manuscript, the authors rely on ‘statistical significance’ to draw their conclusions. The dichotomization of results based on statistical significance using a cut-off value of p<0.05 is strongly discouraged – see for example, one of amy articles on this topic, the paper by Greenland et al 2016. Further, there is no need to draw attention to certain values in your tables by bolding these values based on p-values <0.05.

Greenland, Senn, Rothman, et al. Statistical Tests, p-values, confidence intervals, and power: a guide to mininterpretations. Eur J Epi. 2016; 31(4): 337-350.

Minor edits:

1. line 68 – space between “mostdeadly” (i.e. most deadly)

2. Reference in line 123 should be reference 23 not reference 22.

3. Table 1:

o Months are written inconsistently. For example, October is sometimes written in full, other times as ‘Oct’ and also as ‘Oc’. Please write all months consistently throughout the table.

o Follow up days for study by Ojuawo et al 1998 is missing, so are the age (years) for this study.

o I would suggest using abbreviation ‘N/R’ (Not Reported) for information that wasn’t reported instead of the dash (e.g. when age wasn’t reported). This also applies when standard deviation or range was not reported for age. – use N/R instead of just reporting only the mean.

o Please differentiate reporting of range and interquartile range for age. Range and inter-quartile range are different and should be differentiated in the table.

4. Table 2: Footnote “* Does not add up to 100% since two studies included both age groups and was omitted” should say “Does not add up to 100% since two studies included both age groups and were omitted”

5. The text in lines 161-163 does not directly reflect what is in Table 2. For example, the text says “All 17 RCTs from Africa were conducted among children” but in Table 2 only 16 studies conducted in Africa were in children.

6. Table S3 – the n’s in the first row don’t add up to the total. Same comment for row with data for Africa and Asia (It looks like the numbers in the rows corresponding to Africa and Asia have been swapped).

7. Table 3 – it should be clear that Arteether has not been evaluated in adults, so the effect estimates can not be derived for adults (i.e. in the lower triangle of the figure).

8. Please check statement in line 196 "…descending probability of being the best against quinine.”. The probabilities should be the probability of being the best. However, the effect estimates are compared to quinine – this sentence needs revising.

9. In the Discussion section, all effect estimates should be presented with their corresponding 95% confidence intervals.

Reviewer #2: This paper is a systematic review of all randomised trials in severe malaria comparing the artemisinin derivatives and/or quinine. The main novelty is the use of a network meta-analytic approach which allows for the use of indirect evidence to compare treatments for which there are few data from direct comparisons.

Overall I think the analysis is well done and the accompanying GitHub repository provides all the meta-data and the R scripts used to analyze the data. Everything is therefore easily reproducible and this provides a useful resource for future work.

My main issue is the how the results are summarized and how the evidence was assessed.

Major comments:

First of all, as in most research areas, there are many small randomised trials and only a few large definitive randomised trials. In this case there are only 4 large studies (>500 patients):

-Hien 1996 (n=560)

-Van Hensbroek 1996 (n=576)

-Dondorp 2005 (n=1461)

-Dondorp 2010 (n=5425)

These 4 studies (3 of which were run by the same research group with near identical designs and inclusion criteria) represent >70% of all randomised data. In addition Phu et al (2010, n=370) did the main head-to-head comparison of artesunate versus artemether. It is important to note that the Phu et al trial was done in the same patient population and the same hospital wards as Hien 1996 (in the years following Hien et al, but published much later), thus giving high internal validity to the artesunate vs artemether vs quinine 3 way comparison in adults.

What I can’t understand is how the evidence from these studies, which dominate the available evidence base, can be called “moderate” and “low” quality evidence. As the authors know well, large studies are much less likely to suffer from publication bias, and have greater generalizability. The two largest studies (SEAQUAMAT and AQUAMAT) are both graded as fully “low risk” for the risk of bias (S4 Figure). However, Hien 1996 and Van Hensbroek 1996 both have “some concerns” for the randomization process, but this is very puzzling. Could this be explained? From the papers:

Hien 1996: “Patients were randomly assigned to receive artemether (50 mg per milliliter) or quinine dihydrochloride (250 mg per milliliter) for a minimum of 72 hours. The drugs were issued in packs of 10 identical 3-ml ampules by the Kunming Pharmaceutical Company (Kunming, People’s Republic of China). [..]

The drugs for each patient were placed in a coded sealed envelope, and the envelopes were randomized in blocks of 20. Once a patient was enrolled in the study the envelope was opened. Subsequent analysis of efficacy was on an intention-to-treat basis. Although the ampules, drug volumes, and administration schedules of artemether and quinine were identical, the viscosity and color of the two drugs were slightly different. To maintain blinding, a separate team of nurses, who were not otherwise involved with the care of the study patients, drew up and gave the injections. The drugs were kept in an opaque packet in a locked cabinet during the study.”

Van Hensbroek 1996: “The treatment code for each child was stored in a sealed envelope that was opened after the admission procedure was completed and parental consent had been obtained. The different dose schedules used for artemether and quinine meant that the ward staff members were aware of the patients’ treatment assignments. However, the assessment of neurologic sequelae in survivors was carried out by a doctor who was unaware of the treatment code.”

In both cases the randomisation procedure seems more than adequate! Hien 1996 took special care to maintain blinding (more that what is usually done) and van Hensbroek 1996 was not blinded but the outcomes (other than mortality) were determined by a doctor blinded to assignment. In both cases, this is as close to gold standard procedures as is possible in low resource settings. I see very low risk of bias in both studies for the mortality endpoint.

Thus, when the abstract says “Compared to quinine, artesunate reduced mortality in children (risk ratio (RR), 0.76; 95%CI [0.65 to 0.89], moderate quality), adults (RR, 0.55; 95%CI [0.40 to 0.75], low quality) and in cerebral malaria (RR, 0.72; 95%CI [0.55 to 0.94], moderate quality)” I find this rather hard to reconcile with what i know about these trials!

My second major comment concerns the fact that the analysis does not manage to establish a “batting order” for the three main drugs: artesunate, artemether and quinine. It’s a near certainty that artesunate >> quinine (randomised trial evidence, biological mechanisms etc). It seems pretty clear that artesunate >> artemether (randomised evidence + pharmacokinetics). Although we expect treatment effects to differ between children and adults, we don’t expect the rankings to change. So why not do a full network analysis including randomised trials in children and adults for artesunate, artemether and quinine to determine the relative benefits of each? Again this would be primarily determined by Hien 1996, Hensbroek 1996, Dondorp 2005&2010 and Phu 2010…

Minor comments:

Hypoglycaemia was not stated as an outcome in the earlier section of the Methods but then stated as an outcome later on.

Coma recovery time and other “intermediate” endpoints: estimates will be biased as you need to adjust for mortality as a competing risk. This can only be done with individual patient data.

Fig 2: surely this makes no sense to show thickness in terms of the number of studies? The number of patients in all randomised comparisons is much more important (eg ten studies each with 50 patients, n total is 500, versus 1 study with 5000 patients!)

This is exactly what has happened with artesunate.

Table 1 has some problems.

Hien 1996: the dates are stated in the paper (“Between May 1991 and January 1996, 561 patients were enrolled in the study”)

Van Hensbroek 1996: not IV QN but intramuscular (abstract: We conducted a randomized, unblinded comparison of intramuscular artemether and intramuscular quinine in 576 Gambian children with cerebral malaria)

6. PLOS authors have the option to publish the peer review history of their article (what does this mean?). If published, this will include your full peer review and any attached files.

Reviewer #1: No

Reviewer #2: **Yes: **James Watson

---

## [Author Response · Author response to Decision Letter 0]

30 Dec 2021

ACADEMIC EDITOR 

Journal Requirements:

Response: Supplementary materials were edited to include ‘Fig’ or ‘Table’ as appropriate. 

Response: Corresponding author’s name was added, and initials were also included after the corresponding author’s email. The second group of contributors was also edited to “&JMOM and JH are Joint Senior Authors”.

“Initial part of this review was done as part of Nicholas Nyaaba’s MSc from the University of Oxford which was supported by the Ghana Education Trust Fund (GETFund).”

“The authors received no funding for this work.”

Response: We have removed funding information from the Acknowledgement and Funding Statement has been amended as 

“This study did not directly receive any funding. Initial part of this review was done as part of Nicholas Nyaaba’s MSc from the University of Oxford which was supported by the Ghana Education Trust Fund (GETFund) and the Ghana Armed Forces Medical Services (GAFMS). The funders had no role in study design, data collection and analysis, decision to publish, or preparation of the manuscript.”

This is included in the Cover Letter.

Response: ORCID iD for corresponding author has been created and information has been updated. 

Reviewer #1: 

This systematic review and network meta-analysis aimed to assess the efficacy and safety of the artemisinin derivatives and quinine for treating severe P. falciparum malaria in children and adults by updating previous published Cochrane reviews. I have focused my review on the methodological and statistical aspects of this manuscript.

Response: Thank you very much for the time dedicated to reviewing our manuscript. We acknowledge that the Reviewer raises very important concerns, most of which are as a result of our efforts to manage the word count. We subsequently, amended the manuscript to resolve these concerns.

Major comments:

1. The statistical methods section of this review is lacking many details that are either required for a network meta-analysis or are needed to align with the results presented by the authors. I strongly suggest that the authors consult the paper by Chaimani et al 2017, which provides more details than the PRISMA NMA for conducting a network meta-analysis. The authors should also be aware that the PRISMA guidelines have been updated and PRISMA 2020 (Page 2021) should be used in conjunction with PRISMA NMA.

Chamaini et al. Additional considerations are required when preparing a protocol for a systematic review with multiple interventions. Journal of Clinical Epidemiology. 2017: 83; 65-74.

Page et al. The PRISMA 2020 statement: an updated guideline for reporting systematic reviews. BMJ. 2021; 372:n71.

Response: We have added more subsections with more details to address this concern. In our attempt to manage the word count, we missed many of these details. We have also updated our reference (see reference number 21) and provided a more useful PRISMA table to accompany this manuscript re-submission. 

2. The authors have converted medians (range/inter-quartile range) to mean (Standard deviation) and used this in the main analysis. Median and (range/inter-quartile range) are reported when the data do not have a normal distribution. By converting these data to mean (Standard deviation) the authors are assuming a normal distribution. Converting summary statistics for skewed data to summary staitstics used for normally distributed data doesn’t make sense. The authors should produce a meta-analysis of ratios of geometric means when skewed data are reported on the raw scale. Further details of methods used to produce a meta-analysis of skewed data are available in Higgins et al 2008.

Higgins, White and Anzures-Cabrera. Meta-analysis of skewed data: combining results reported on log-transformed or raw scales. Statistics in Medicine. 2008; 27;6072-92.

Response: Thank you. Please, notice that the methods we applied are those recommended in the Cochrane Handbook of Systematic Reviews. Specifically, the methods by Wan et al has been recommended because it incorporates sample size compared to older methods (Hozo et al and Blands). Cognizant of the limitations mentioned, we considered excluding these studies in which the median was reported, and we did not find them to be very influential on the main results. Furthermore, our main outcome was not a continuous variable. We therefore believe this is unlikely to have major influence on the results and conclusions of our study. We also believe that it was an opportunity to review this method on real live data.

Hozo SP, Djulbegovic B, Hozo I: Estimating the mean and variance fromthemedian, range, and the size of a sample. BMCMed Res Methodol 2005, 5:13.

Bland M: Estimating mean and standard deviation from the sample size, three quartiles, minimum, and maximum. International Journal of Statistics in Medical Research, in press. 2014.

3. The methods section would benefit from more subsection, including a section that explicitly states that inclusion/exclusion criteria for the study.

Response: Thank you. We have now added an inclusion and exclusion subsection in the Methods (see lines 104-112)

4. The eligibility criteria for a network meta-analysis should clearly consider the transitivity assumption. The transitivity assumption is the fundamental assumption underlying the validity of a network meta-analysis. The authors have not considered the transitivity assumption for their network meta-analysis, which is a fundamental error when undertaking a network meta-analysis.

Response: Thank you. We assessed transitivity at two levels to show the distribution of studies across effect modifiers; Table 2 at the study level and S2 Table at the comparison level. We have added a subsection in the Methods/Results? to clearly address this concern (lines 138-142). We also provided explanations in the Result session (lines 208-211).

“Sixteen out of 17 RCTs from Africa were conducted among children and 11 out of 12 RCTs conducted among adults were in Asia. Out of the 12 cerebral malaria only RCTs, eight were from Africa and four from Asia.”

 In the Discussion (lines 414-418) we also stated that 

“We detected an unequal distribution of RCTs in age groups among Asian and African population; sixteen out of 17 RCTs from Africa were conducted among children and 11 out of 12 RCTs conducted among adults were in Asia. The earliest systematic reviews found that artemether was more effective among Asian adults than in African children [13,18]. This has partly motivated subsequent RCTs along these subgroups”

5. The statistical methods section needs to provide an explanation of how the network diagram is created (i.e. what do the size of the nodes and thickness of the edges represent).

Response: Thank you. We have added a description in the Methods Section of the how the network diagrams were created (see lines 162-165).

6. The authors conduct a meta-analysis of rare events (i.e. analysis of serious adverse events and neurological sequalae) but haven’t considered methods for the meta-analysis of rare events or how they handled studies with no events in one or more arms. Further details are provided in the Cochrane handbook https://training.cochrane.org/handbook/archive/v6.1/chapter-10#section-10-4-4.

Response: Thank you. We have amended our analyses of rare events. We analysed Hypoglycemia events using Mantel-Haenzel method, and Neurological Sequela events and ECG abnormalities with Peto’s method. We have stated this in the Methods Section (see lines 155-158). Please note that the results compared to our initial analyses were similar.

7. In figure 1, 2 studies were excluded because the outcomes relevant to this review were not reported. Did the review authors attempt to contact the original study authors to obtain the necessary outcome data?

Response: Thank you. Two studies were excluded because they did not measure the outcomes of interest, not because they measured them but did not report them. We realized we may not have made this clear enough and we have therefore clarified this in S2 and Fig 1. Byakika-Kibwika et al only measured days to parasite clearance, and Barnes et al recorded proportion of patients with asexual parasitaemia of less than 60% after 12 hours.

8. In the results section the authors state: "A traditional meta-analysis conducted as shown in S11, found that artemether…" but details of this analysis are not provided in the methods section.

Response: Thank you. We have amended this in the Methods Section (see lines 156-159).

9. Methods used to estimate the between-study heterogeneity and calculation of the 95% CIs needs to be explicitly stated in the methods section of the review.

Response: Thank you. We have amended this in the Methods Section (see lines 168-169).

10. Subgroup and sensitivity analyses need to be explicitly stated in the statistical methods section.

Response: Thank you. We have made this clearer by providing an Additional analyses subsection (see lines 178-183). 

11. Throughout the manuscript, the authors rely on ‘statistical significance’ to draw their conclusions. The dichotomization of results based on statistical significance using a cut-off value of p<0.05 is strongly discouraged – see for example, one of amy articles on this topic, the paper by Greenland et al 2016. Further, there is no need to draw attention to certain values in your tables by bolding these values based on p-values <0.05.

Greenland, Senn, Rothman, et al. Statistical Tests, p-values, confidence intervals, and power: a guide to mininterpretations. Eur J Epi. 2016; 31(4): 337-350.

Response: Thank you and apologies for previously putting too much emphasis on statistical significance. Due to the large number of outcomes and drugs being compared, we thought it was useful to highlight significant results in the tables. We have now amended the manuscript throughout to lay less emphasis on statistically significant results. (see lines 257-264,308-310, 367-369, etc)

Minor edits:

1. line 68 – space between “mostdeadly” (i.e. most deadly)

Response: Thank you. This has been corrected.

2. Reference in line 123 should be reference 23 not reference 22.

Response: Thank you. This has been corrected.

3. Table 1:

o Months are written inconsistently. For example, October is sometimes written in full, other times as ‘Oct’ and also as ‘Oc’. Please write all months consistently throughout the table.

Response: Thank you. This has been corrected.

o Follow up days for study by Ojuawo et al 1998 is missing, so are the age (years) for this study.

Response: Thank you. This has been corrected.

o I would suggest using abbreviation ‘N/R’ (Not Reported) for information that wasn’t reported instead of the dash (e.g. when age wasn’t reported). This also applies when standard deviation or range was not reported for age. – use N/R instead of just reporting only the mean.

Response: Thank you. This has been amended as suggested.

o Please differentiate reporting of range and interquartile range for age. Range and inter-quartile range are different and should be differentiated in the table.

Response: Thank you. This has been clarified.

4. Table 2: Footnote “* Does not add up to 100% since two studies included both age groups and was omitted” should say “Does not add up to 100% since two studies included both age groups and were omitted”

Response: Thank you. This has been corrected.

5. The text in lines 161-163 does not directly reflect what is in Table 2. For example, the text says “All 17 RCTs from Africa were conducted among children” but in Table 2 only 16 studies conducted in Africa were in children.

Response: Thank you. We have amended this (see lines 208-210) to read “Sixteen out of 17 RCTs from Africa were conducted among children and 11 out of 12 RCTs conducted among adults were in Asia. Out of the 12 cerebral malaria only RCTs, eight were from Africa and four from Asia.”

6. Table S3 – the n’s in the first row don’t add up to the total. Same comment for row with data for Africa and Asia (It looks like the numbers in the rows corresponding to Africa and Asia have been swapped).

Response: Thank you. S3 Table shows the frequency distribution of pairwise comparison across effect modifiers. The total number of pairwise comparisons are 41. We have cross checked data on S3 Table and they are all correct. To clarify we have added a footnote to this supplementary table.

7. Table 3 – it should be clear that Arteether has not been evaluated in adults, so the effect estimates can not be derived for adults (i.e. in the lower triangle of the figure).

Response: Thank you. This has been clarified in the results section (see lines 221 and 227) and the footnote of Table 3 (see line 268)

8. Please check statement in line 196 "…descending probability of being the best against quinine.”. The probabilities should be the probability of being the best. However, the effect estimates are compared to quinine – this sentence needs revising.

Response: Thank you. This has been amended (see line 230)

9. In the Discussion section, all effect estimates should be presented with their corresponding 95% confidence intervals.

Response: Thank you. We have amended this in the Discussion section (see lines 359-362).

 

Reviewer #2

This paper is a systematic review of all randomised trials in severe malaria comparing the artemisinin derivatives and/or quinine. The main novelty is the use of a network meta-analytic approach which allows for the use of indirect evidence to compare treatments for which there are few data from direct comparisons.

Overall I think the analysis is well done and the accompanying GitHub repository provides all the meta-data and the R scripts used to analyze the data. Everything is therefore easily reproducible and this provides a useful resource for future work.

My main issue is the how the results are summarized and how the evidence was assessed.

Major comments:

First of all, as in most research areas, there are many small randomised trials and only a few large definitive randomised trials. In this case there are only 4 large studies (>500 patients):

-Hien 1996 (n=560)

-Van Hensbroek 1996 (n=576)

-Dondorp 2005 (n=1461)

-Dondorp 2010 (n=5425)

These 4 studies (3 of which were run by the same research group with near identical designs and inclusion criteria) represent >70% of all randomised data. In addition Phu et al (2010, n=370) did the main head-to-head comparison of artesunate versus artemether. It is important to note that the Phu et al trial was done in the same patient population and the same hospital wards as Hien 1996 (in the years following Hien et al, but published much later), thus giving high internal validity to the artesunate vs artemether vs quinine 3 way comparison in adults.

What I can’t understand is how the evidence from these studies, which dominate the available evidence base, can be called “moderate” and “low” quality evidence. As the authors know well, large studies are much less likely to suffer from publication bias, and have greater generalizability. The two largest studies (SEAQUAMAT and AQUAMAT) are both graded as fully “low risk” for the risk of bias (S4 Figure). However, Hien 1996 and Van Hensbroek 1996 both have “some concerns” for the randomization process, but this is very puzzling. Could this be explained? From the papers:

Hien 1996: “Patients were randomly assigned to receive artemether (50 mg per milliliter) or quinine dihydrochloride (250 mg per milliliter) for a minimum of 72 hours. The drugs were issued in packs of 10 identical 3-ml ampules by the Kunming Pharmaceutical Company (Kunming, People’s Republic of China). [..]

The drugs for each patient were placed in a coded sealed envelope, and the envelopes were randomized in blocks of 20. Once a patient was enrolled in the study the envelope was opened. Subsequent analysis of efficacy was on an intention-to-treat basis. Although the ampules, drug volumes, and administration schedules of artemether and quinine were identical, the viscosity and color of the two drugs were slightly different. To maintain blinding, a separate team of nurses, who were not otherwise involved with the care of the study patients, drew up and gave the injections. The drugs were kept in an opaque packet in a locked cabinet during the study.”

Van Hensbroek 1996: “The treatment code for each child was stored in a sealed envelope that was opened after the admission procedure was completed and parental consent had been obtained. The different dose schedules used for artemether and quinine meant that the ward staff members were aware of the patients’ treatment assignments. However, the assessment of neurologic sequelae in survivors was carried out by a doctor who was unaware of the treatment code.”

In both cases the randomisation procedure seems more than adequate! Hien 1996 took special care to maintain blinding (more that what is usually done) and van Hensbroek 1996 was not blinded but the outcomes (other than mortality) were determined by a doctor blinded to assignment. In both cases, this is as close to gold standard procedures as is possible in low resource settings. I see very low risk of bias in both studies for the mortality endpoint.

Thus, when the abstract says “Compared to quinine, artesunate reduced mortality in children (risk ratio (RR), 0.76; 95%CI [0.65 to 0.89], moderate quality), adults (RR, 0.55; 95%CI [0.40 to 0.75], low quality) and in cerebral malaria (RR, 0.72; 95%CI [0.55 to 0.94], moderate quality)” I find this rather hard to reconcile with what i know about these trials!

Response: Thank you. The quality of evidence produced, was assessed using the Confidence in Network Meta-analysis (CINeMA) approach which classifies evidence as high, moderate, low or very low quality based on the within-study bias, heterogeneity, reporting bias, imprecision, indirectness and incoherence. 

In CINeMA, the within-study-bias represents inadequacies in the conduct of research that can lead to deviation from the truth. This is measured using the assessment of risk of bias tool. The Reviewer raises important concerns which only relates to the within-study-bias. CINeMA considers these concerns by combining the studies’ contributions with the risk of bias judgments to evaluate within-study bias for each estimate from a network meta-analysis. It uses the percentage contribution of each study and then computes the percentage contribution from studies judged to be at low, moderate, and high risk of bias. This is generated from the CINeMA webpage (https://cinema.ispm.unibe.ch/). Please, refer to this website and the documentation within that site for more details regarding the methodology.

We have adjusted our judgements on within-study-bias in the CINeMA webpage to accommodate the Reviewers concerns and have ammended this in S14. This is seen in the Contribution Matrix presented below. However, we have maintained our judgement on Reporting Bias because both Egger’s Test and Funnel plot indicates publication bias. Consequently, Artesunate vs Quinine for both age groups, are assigned Moderate Quality. We have amended S14 to capture details of the grading done with CINeMA.

Risk of Bias for Each Comparison in the Adult analyses

Risk of Bias for Each Comparison in the Children Analyses

 These figures represent the proportion of low, moderate or high risk of bias studies contributing to each comparison. 

My second major comment concerns the fact that the analysis does not manage to establish a “batting order” for the three main drugs: artesunate, artemether and quinine. It’s a near certainty that artesunate >> quinine (randomised trial evidence, biological mechanisms etc). It seems pretty clear that artesunate >> artemether (randomised evidence + pharmacokinetics). Although we expect treatment effects to differ between children and adults, we don’t expect the rankings to change. So why not do a full network analysis including randomised trials in children and adults for artesunate, artemether and quinine to determine the relative benefits of each? Again this would be primarily determined by Hien 1996, Hensbroek 1996, Dondorp 2005&2010 and Phu 2010…

Response: Thank you. We conducted sensitivity analyses lumping all studies regardless of age groups, as we indicated in our protocol. The results were similar. However, we did not add this to the Manuscript because we wanted to cut down on the word count. We have now amended the manuscript by mentioning this in the methods section (see line 183) and result section (lines 343-345) and provided the full results in S15 Table. 

Response: Thank you. As for the ranking of treatments, we considered the results from this review including all randomized clinical trials (regardless of their sample size) comparing these drugs, and considering several outcomes, the evidence does not support that either artemisinin derivatives is superior to the others for all outcomes consistently. Therefore, the ranking of treatments was not consistent across outcomes. This is one of the main findings of this review. In this review, we did not systematically look at pharmacokinetics or studies detailing biological mechanisms, instead we looked at randomized controlled trials, which provide the best evidence for determining treatments effectiveness. 

Minor comments:

Hypoglycaemia was not stated as an outcome in the earlier section of the Methods but then stated as an outcome later on.

Response: Thank you. We used adverse events to capture Hypoglycaemia and ECG abnormalities. We have amended this in the method by stating these adverse events (see lines 119-122). 

Coma recovery time and other “intermediate” endpoints: estimates will be biased as you need to adjust for mortality as a competing risk. This can only be done with individual patient data.

Response: Thank you. We acknowledge the Reviewers concern that there is a more appropriate method for analyzing Coma Recovery Time using Individual Patient Data. However, that kind of data was not available so we used methods also used by previous Cochrane Systematic Reviews.

Fig 2: surely this makes no sense to show thickness in terms of the number of studies? The number of patients in all randomised comparisons is much more important (eg ten studies each with 50 patients, n total is 500, versus 1 study with 5000 patients!)

This is exactly what has happened with artesunate.

Response: Thank you. We acknowledge the Reviewers concern that it is important to represent the number of patients. As shown in the Figure 2 and legend, the blue nodes are proportional to the number of participants allocated to that drug. The thickness of the black edges is proportional to the number of studies comparing the drugs on each side of the edge.

Table 1 has some problems.

Hien 1996: the dates are stated in the paper (“Between May 1991 and January 1996, 561 patients were enrolled in the study”)

Van Hensbroek 1996: not IV QN but intramuscular (abstract: We conducted a randomized, unblinded comparison of intramuscular artemether and intramuscular quinine in 576 Gambian children with cerebral malaria)

Response: Thank you. We have amended this in Table 1.

6. PLOS authors have the option to publish the peer review history of their article (what does this mean?). If published, this will include your full peer review and any attached files.

Do you want your identity to be public for this peer review? For information about this choice, including consent withdrawal, please see our Privacy Policy.

Reviewer #1: No

Reviewer #2: Yes: James Watson

---

## [Decision Letter · Decision Letter 1]

7 Mar 2022

PONE-D-21-29782R1Comparative efficacy and safety of the Artemisinin Derivatives compared to Quinine for treating Severe Malaria in Children and Adults: a Systematic Update of Literature and Network Meta-analysisPLOS ONE

Dear Dr. Nyaaba,

Thank you for submitting your manuscript to PLOS ONE. After careful consideration, we feel that it has merit but does not fully meet PLOS ONE’s publication criteria as it currently stands. Therefore, we invite you to submit a revised version of the manuscript that addresses the points raised during the review process.

We look forward to receiving your revised manuscript.

Kind regards,

Benedikt Ley, PhD

Academic Editor

PLOS ONE

Reviewers' comments:

Reviewer's Responses to Questions

**Comments to the Author**

1. If the authors have adequately addressed your comments raised in a previous round of review and you feel that this manuscript is now acceptable for publication, you may indicate that here to bypass the “Comments to the Author” section, enter your conflict of interest statement in the “Confidential to Editor” section, and submit your "Accept" recommendation.

Reviewer #1: (No Response)

Reviewer #2: (No Response)

2. Is the manuscript technically sound, and do the data support the conclusions?

Reviewer #1: Partly

Reviewer #2: Yes

3. Has the statistical analysis been performed appropriately and rigorously? 

Reviewer #1: Yes

Reviewer #2: Yes

4. Have the authors made all data underlying the findings in their manuscript fully available?

Reviewer #1: Yes

Reviewer #2: Yes

5. Is the manuscript presented in an intelligible fashion and written in standard English?

Reviewer #1: Yes

Reviewer #2: Yes

6. Review Comments to the Author

Reviewer #1: Further comments specific to the current manuscript and the authors response to the reviewers’ comments are provided below:

1. Transitivity requires that the requirements below are satisfied:

a. The treatments included in the meta-analysis are considered to be ‘jointly randomizable’,

b. The different trials are similar enough to be combined,

c. The characteristics associated with the effect of the treatments are similar across the included trials (Salanti 2012 and Chaimani et al 2021).

Requirements a and b need to considered at the planning stage of the network meta-analysis, and requirement c is assessed after the data from each of the included trials has been collected. The authors have only assessed requirement c and have not considered requirements a and b.

In addition, in response to the other reviewer, the authors have presented the results all studies (i.e., combining trials in adults and children together). However, this again, is only justifiable if the transitivity assumption is adhered to (see requirements a above). If the treatments given to children and to adults can not be considered to be jointly randomizable then it does not make sense to present these results pooled together.

2. The authors have claimed that they followed the methods recommended in the Cochrane Handbook for analysis of skewed data. However, the Cochrane handbook states that “analyses based on means are appropriate for data that are at least approximately normally distributed… Review authors should consider the possibility and implications of skewed data when analysing continuous outcomes” (section 10.5.3 https://training.cochrane.org/handbook/current/chapter-10#section-10-5-3). Further, they recommend that “Collection of appropriate data summaries from the trialists, or acquisition of individual patient data, is currently the approach of choice.” (section 10.5.3 https://training.cochrane.org/handbook/current/chapter-10#section-10-5-3).

3. I agree with the other reviewer that showing the thickness of the edges in the network map as number of studies doesn’t make sense. I would strongly suggest that the authors consider changing this to correspond to the precision of the estimate instead of the number of studies.

4. The authors should make it clear which analyses were added post hoc.

5. The link to the GitHub code does not work. I received a ‘404’ error message when I tried to access it.

6. The netheat plot and forest plot of direct and indirect estimates are not presented for the primary outcome (mortality). Please explain or provide.

7. The estimates reported in the manuscript in lines 322-325 are RRs but are referencing the figure S11, which shows odds ratios. This is very confusing. This applies to the estimates in lines 327-330 and Figure S12.

8. In lines 322-325, it should be clear how many studies are contributing to the pooled estimates. Most of the comparisons in Figure S11 are from 1 study.

9. In the text, the authors state that ‘Figure S11’ is a network forest plot but examination of Figure S11 looks like a pairwise Forest plot. Please clarify. Further, if the estimates are from a network meta-analysis it needs to be clearer how Peto method was invoked in the netmeta package in R.

10. For Figure 5, it needs to be clear what is being plotted here – is it just the estimates for artemisinin vs quinine (as reads in the manuscript lines 347/348)? If so, where are the forest plots for the remaining analyses?

11. Minor comments:

a. There are some grammatical errors:

i. For example, lines 126/127 “For binary outcomes, the number of participants 127 experiencing the event and numbers assessed in each randomised group was recorded.” Should be “For binary outcomes, the number of participants 127 experiencing the event and numbers assessed in each randomised group were recorded.”

ii. Similarly, line 152 “Risk ratios was pooled… “ should be “Risk Ratios were pooled…”

iii. Netheat is sometimes spelled as two words ‘net heat’ and other times as one word ‘netheat’, please make it consistent throughout the manuscript (including in the figures).

b. Lines 202-203, please add references consistently – i.e., you have provided the reference for the trial completed in South Pacific but have not provided the references for the trials completed in Asia or Africa. This applies to all references to trials in this paragraph and throughout the manuscript.

c. The age group for the study by Haroon et la (ref #39) in Table 1 is missing.

d. Lines 205-206 state that “Twelve RCTS were among participants with only cerebral malaria…” but I count 13 in Table 1.

e. In table 3, it needs to be clear in the footnote which test was used to generate the p-value for heterogeneity.

f. In Table 3, it needs to be clear in the footnote how inconsistency was assessed.

g. In Table 3, it needs to be clear what the difference is between total variability and heterogeneity, and how they are calculated.

12. A minor comment to the authors for future response to reviewers – I find it helpful if the response indicates the specific changes that are made and corresponding line numbers. This makes it easier as a review to identify exactly how the authors have addressed the concerns raised.

References

Salanti G. Indirect and mixed-treatment comparison, network, or multiple-treatments meta-analysis: many names, many benefits, many concerns for the next generation evidence synthesis tool. Research Synthesis Methods 2012; 3: 80–97.

Chaimani A, Caldwell DM, Li T, Higgins JPT, Salanti G. Chapter 11: Undertaking network meta-analyses. In: Higgins JPT, Thomas J, Chandler J, Cumpston M, Li T, Page MJ, Welch VA (editors). Cochrane Handbook for Systematic Reviews of Interventions version 6.2 (updated February 2021). Cochrane, 2021. Available from www.training.cochrane.org/handbook.

Deeks JJ, Higgins JPT, Altman DG (editors). Chapter 10: Analysing data and undertaking meta-analyses. In: Higgins JPT, Thomas J, Chandler J, Cumpston M, Li T, Page MJ, Welch VA (editors). Cochrane Handbook for Systematic Reviews of Interventions version 6.3 (updated February 2022). Cochrane, 2022. Available from www.training.cochrane.org/handbook.

Reviewer #2: Some final comments I would like to see addressed before publication:

Introduction, paragraph 2: please make it clear here that you are focusing on severe falciparum malaria. Falciparum and knowlesi are the two parasites which sequester in the microvasculature thus causing severe disease. Severe vivax is very different in terms of presentation and the exact definition is debated.

In addition severe falciparum is diagnosed from evidence of *asexual* stages in circulation.

Results: surely Figure 2 is wrong? In panel A the circle for ASU (artesunate) is very small, whereas ~3000 patients were randomised to ASU? Maybe I am missing something, but I don't quite understand this Figure.

Discussion: the authors write: “All artemisinin drugs except artemether may increase neurological sequela events in the first three to seven days of treatment, compared to quinine. Most of the neurological sequela events resolved during follow up, which is similar to findings of other studies [5,40,53].”

This should be caveated by the bias I mentioned in my first review: death and coma recovery are competing events: a drug that saves lives in patients who would have died with a different drug could thus result in longer coma recovery times (those patients would have died!). The authors do not have access to IPD and thus the effect estimates are likely to be biased.

In response to reviewer comments, the authors wrote:

“We have adjusted our judgements on within-study-bias in the CINeMA webpage to accommodate the Reviewers concerns and have ammended this in S14. This is seen in the Contribution Matrix presented below.”

However when I look at the Sup Material page 5, I still see a yellow "!" for Hien 1996 and Van Hensbroek 1996. Have the authors changed their assessment of “concerns about the randomization process” for these two studies? I strongly believe that the randomization process was more than adequate in these two studies as explained in detail in the first review.

The authors wrote:

“This is one of the main findings of this review. In this review, we did not systematically look at pharmacokinetics or studies detailing biological mechanisms, instead we looked at randomized controlled trials, which provide the best evidence for determining treatments effectiveness.”

For death - the main outcome we care about in the treatment of severe malaria (a disease with 10-30% mortality!) - it is pretty clear that artesunate is better than quinine. I agree that RCTs are the best evidence here.

For parasite clearance, this is also very clear: this is the main surrogate marker of drug activity in vivo.

Neurological sequelae have not be very well documented in these trials as it is very hard to consistently follow patients up after hospital discharge (Hien 1996 is only large trial to have done proper follow-up). As I have stated before, coma recovery times based on aggregated data are biased. So, no, I do not think that a main finding is a lack of consistency. A main finding is lack of data or lack of access to IPD for these outcomes. IPD meta-analysis would be much better to estimate the effects for these secondary outcomes.

7. PLOS authors have the option to publish the peer review history of their article (what does this mean?). If published, this will include your full peer review and any attached files.

Reviewer #1: No

Reviewer #2: **Yes: **James Watson

---

## [Author Response · Author response to Decision Letter 1]

21 Apr 2022

Reviewer #1

Reviewer #1: Further comments specific to the current manuscript and the authors response to the reviewers’ comments are provided below:

1. Transitivity requires that the requirements below are satisfied:

a. The treatments included in the meta-analysis are considered to be ‘jointly randomizable’,

b. The different trials are similar enough to be combined,

Response: Thank you. We had considered this assumption but had not included details in order to keep within the word count. We have now added an ‘Interventions’ section and more details to the “Inclusion and Exclusion Criteria” section to show that the treatments and participants were ‘jointly randomizable’. (Line 105-116, Line 125-128, Line 155-157)

“Interventions

This review covered parenteral and rectal artemisinin drugs administered during the critical phase of treating P. falciparum severe malaria until oral antimalarial can be tolerated by the patient. Parenteral interventions cover both IV and IM interventions. Just as IV and IM artesunate are considered to be similar5, IV and IM quinine do not have significantly different therapeutic benefits in treating severe malaria.65,66 Rectal artemisinin has been compared to parenteral quinine and artesunate in previous studies and has been found to be equally effective in treating severe malaria.48–50 Therefore, both parenteral and rectal interventions were pooled together in the analyses. However adverse events were analyzed and interpreted with the assumption that local reactions will differ with route of administration. Different dosages for the interventions within recommended ranges were combined into a single node, therefore, trials comparing the same drug at different doses were excluded.”

 “The criteria for inclusion ensured that all RCTs included in this review compared the artemisinin derivatives and quinine for the treatment of P. falciparum among adults and children, in head to head comparisons. Children were considered as those aged < 15 years. Trials which included pregnant women were excluded.”

“The interventions in the study were similar in all comparisons. Also, the participants in both adult and children analyses were similar in all studies and could have been assigned to any of the treatments as indicated above, therefore, meeting the joint randomisability requirement for NMA.”

c. The characteristics associated with the effect of the treatments are similar across the included trials (Salanti 2012 and Chaimani et al 2021).

Requirements a and b need to considered at the planning stage of the network meta-analysis, and requirement c is assessed after the data from each of the included trials has been collected. The authors have only assessed requirement c and have not considered requirements a and b.

In addition, in response to the other reviewer, the authors have presented the results all studies (i.e., combining trials in adults and children together). However, this again, is only justifiable if the transitivity assumption is adhered to (see requirements a above). If the treatments given to children and to adults cannot be considered to be jointly randomizable then it does not make sense to present these results pooled together.

Response: Thank you. We acknowledge the Reviewer’s concern so we took steps to address this in our analyses. Separate analyses were conducted for adults and children to reduce clinical inconsistency and heterogeneity, for the main analyses. However, we only pooled all studies regardless of age groups in sensitivity analyses. We have added a sentence to the “Assessment of Transitivity” section to clarify this. (Line 155-157, Line 169-170)

“The interventions in the study were similar in all comparisons. Also, the participants in both adult and children analyses were similar in all studies and could have been assigned to any of the treatments as indicated above, therefore, meeting the joint randomisability requirement for NMA.”

“Separate analyses were conducted for adults and children to reduce clinical inconsistency and heterogeneity, as well as meet the transitivity requirement”

2. The authors have claimed that they followed the methods recommended in the Cochrane Handbook for analysis of skewed data. However, the Cochrane handbook states that “analyses based on means are appropriate for data that are at least approximately normally distributed… Review authors should consider the possibility and implications of skewed data when analysing continuous outcomes” (section 10.5.3 https://training.cochrane.org/handbook/current/chapter-10#section-10-5-3). Further, they recommend that “Collection of appropriate data summaries from the trialists, or acquisition of individual patient data, is currently the approach of choice.” (section 10.5.3 https://training.cochrane.org/handbook/current/chapter-10#section-10-5-3).

Response: Thank you. We acknowledge the Reviewer’s concern. Based on the recommendations of the Reviewer and the Cochrane Handbook for Systematic Reviews we have compared continuous outcomes using mean difference for the main analyses and have only added studies in which we made conversions in the sensitivity analyses. (Line 147-149, Table 3, 300-335)

“Where medians and range or interquartile range were reported instead of the means and standard deviations, the latter were estimated using Wang’s method [28]. These estimations were only used in sensitivity analyses.”

3. I agree with the other reviewer that showing the thickness of the edges in the network map as number of studies doesn’t make sense. I would strongly suggest that the authors consider changing this to correspond to the precision of the estimate instead of the number of studies.

Response: Thank you. We acknowledge the Reviewer’s concern that it is important to represent the number of patients. As shown in the Methods (Line 182-184) nad Figure 2 legend (Line 266-268), the blue nodes are proportional to the number of participants allocated to that drug. The thickness of the black edges is proportional to the number of studies comparing the drugs on each side of the edge. We have followed the convention for network graphs which addresses the Reviewers’ concern, and illustrates both precision (with the size the nodes) and then the number of studies with the edges. 

“The bigger the size of the nodes, the greater the pooled sample size in the treatment. The thicker the edges, the greater the number of studies comparing the treatment.”

“The blue nodes are proportional to the number of participants allocated to that drug. The thickness of the black edges is proportional to the number of studies comparing the drugs on each side of the edge.”

Chaimani A, Higgins JPT, Mavridis D, Spyridonos P, Salanti G (2013) Graphical Tools for Network Meta-Analysis in STATA. PLoS ONE 8(10): e76654. https://doi.org/10.1371/journal.pone.0076654. https://journals.plos.org/plosone/article?id=10.1371/journal.pone.0076654

4. The authors should make it clear which analyses were added post hoc.

Response: Thank you. We confirm that all the analyses were pre-specified as detailed in the protocol. 

Nyaaba N, Ordonez-Mena J, Hirst J. What are the comparative efficacy and safety of the artemisinin derivatives for treating severe malaria? [Internet]. PROSPERO- International prospective register of systematic reviews; 2020.

https://www.crd.york.ac.uk/prospero/display_record.php?ID=CRD42020218190

5. The link to the GitHub code does not work. I received a ‘404’ error message when I tried to access it.

Response: Thank you. Apologies. This error has been fixed. Github Account

6. The netheat plot and forest plot of direct and indirect estimates are not presented for the primary outcome (mortality). Please explain or provide. 

Response: Thank you. We did not provide netheat plots and forest plots for mortality because the Q and I2 statistics did not indicate any evidence of substantial variability in the effect sizes for mortality. We only used the netheat plots, forest plots of the direct and indirect evidence, and net splitting methods when there was substantial evidence of inconsistency in the network for a particular outcome. We have now made this clear in the Methods section. (Line 193-195)

“Analyses which recorded substantial inconsistencies were further investigated to identify hotspots using net-splitting and design by treatment methods as appropriate, and illustrated with the net-splitting and net heat plots respectively [36,37].”

7. The estimates reported in the manuscript in lines 322-325 are RRs but are referencing the figure S11, which shows odds ratios. This is very confusing. This applies to the estimates in lines 327-330 and Figure S12.

Response: Thank you. We converted the ORs from S9 and S10 B into RRs for easy interpretation and consistency. We have now reported both ORs and RRs for these results as indicated in the Methods (Lines 178-179) and Result sections (Lines 338-342, 350-353).

“ We however interpreted these results using RR by calculation from the Odds Ratios (OR) using the assumed comparator risk which was arbitrarily chosen as the median comparator risk [19].”

“We have converted ORs in S9 and S10 B to RRs for easy interpretation but the overall direction on the forest plots remain the same. Traditional meta-analysis pooling 10 RCTs showed that artemether (OR, 0·87; 95%CI [0·55 to 1·37], RR, 0·87; 95%CI [0·56 to 1·34]) may reduce neurological sequela events compared to quinine with mild heterogeneity (I2=28%, p=0·22). Results can be seen in S9 Fig. 

“A traditional meta-analysis conducted as shown in S10 B, found that artemether (OR, 1·72; 95%CI [0·96 to 3·05], RR, 1·65; 95%CI [0·96 to 2·75]) may increase the number of electrocardiogram abnormalities compared to quinine, and this was associated with moderate heterogeneity (I2=69%, p=0·02).”

8. In lines 322-325, it should be clear how many studies are contributing to the pooled estimates. Most of the comparisons in Figure S11 are from 1 study.

Response: Thank you. We have included this in Line 340.

“Traditional meta-analysis pooling 10 RCTs showed that artemether (RR, 0·87; 95%CI [0·56 to 1.34]) may reduce neurological sequela events compared to quinine with mild heterogeneity (I2=28%, p=0·22). Results can be seen in S 9 Fig.” 

9. In the text, the authors state that ‘Figure S11’ is a network forest plot but examination of Figure S11 looks like a pairwise Forest plot. Please clarify. Further, if the estimates are from a network meta-analysis it needs to be clearer how Peto method was invoked in the netmeta package in R.

Response: Thank you. We have clarified this; S11( now S9) and S10 B are traditional Meta-analysis with Peto’s Method while S10 A is network meta-analysis with Mantel Haenszel method. (Line 338, 345)

10. For Figure 5, it needs to be clear what is being plotted here – is it just the estimates for artemisinin vs quinine (as reads in the manuscript lines 347/348)? If so, where are the forest plots for the remaining analyses?

Response: Thank you. The comparison adjusted funnel plot represents all the artemisinin drugs. We have clarified this in Line 369-370.

“Smaller studies were more likely to produce a beneficial effect in favour of all the artemisinin drugs.”

11. Minor comments:

a. There are some grammatical errors:

i. For example, lines 126/127 “For binary outcomes, the number of participants 127 experiencing the event and numbers assessed in each randomised group was recorded.” Should be “For binary outcomes, the number of participants 127 experiencing the event and numbers assessed in each randomised group were recorded.”

ii. Similarly, line 152 “Risk ratios was pooled… “ should be “Risk Ratios were pooled…”

iii. Netheat is sometimes spelled as two words ‘net heat’ and other times as one word ‘netheat’, please make it consistent throughout the manuscript (including in the figures).

Response: Thank you. These have been corrected. (Line 143,172). It has now been changed to “net heat” plots.

b. Lines 202-203, please add references consistently – i.e., you have provided the reference for the trial completed in South Pacific but have not provided the references for the trials completed in Asia or Africa. This applies to all references to trials in this paragraph and throughout the manuscript.

Response: Thank you. This has been corrected. We have provided a general reference to Table 1 in Line 225.

c. The age group for the study by Haroon et la (ref #39) in Table 1 is missing.

Response: Thank you. We apologise. This has been inserted. 

d. Lines 205-206 state that “Twelve RCTS were among participants with only cerebral malaria…” but I count 13 in Table 1.

Response: Thank you. We have corrected this. Olumese et al was incorrectly entered as cerebral malaria.

e. In table 3, it needs to be clear in the footnote which test was used to generate the p-value for heterogeneity.

Response: Thank you. We have added a statement to indicate this. Line 297-298.

“The DerSimonian and Laird method was used to estimate the between-study variance, and the Jacksons method to estimate its confidence intervals [35,31]

P-values were two-sided with a significance level of 0.05.”

f. In Table 3, it needs to be clear in the footnote how inconsistency was assessed.

Response: Thank you. We have added a sentence to show this. Line (299)

 “Cochran Q method was used to assess the total variability and then decomposed into heterogeneity and inconsistency [36].”

g. In Table 3, it needs to be clear what the difference is between total variability and heterogeneity, and how they are calculated.

Response: Thank you. We have added a sentence in the “Assessment of Heterogeneity and Inconsistency” section. (Line 190-192)

“The variability among the individual study estimates as compared to the network estimates was represented by the total variability, using the Cochran Q method [32]. This was then broken down into within-design (heterogeneity) and between-design (inconsistency).”

12. A minor comment to the authors for future response to reviewers – I find it helpful if the response indicates the specific changes that are made and corresponding line numbers. This makes it easier as a review to identify exactly how the authors have addressed the concerns raised.

Response: Thank you. We have added line numbers when responding to the reviewers’ comments.

References

Salanti G. Indirect and mixed-treatment comparison, network, or multiple-treatments meta-analysis: many names, many benefits, many concerns for the next generation evidence synthesis tool. Research Synthesis Methods 2012; 3: 80–97.

Chaimani A, Caldwell DM, Li T, Higgins JPT, Salanti G. Chapter 11: Undertaking network meta-analyses. In: Higgins JPT, Thomas J, Chandler J, Cumpston M, Li T, Page MJ, Welch VA (editors). Cochrane Handbook for Systematic Reviews of Interventions version 6.2 (updated February 2021). Cochrane, 2021. Available from www.training.cochrane.org/handbook.

Deeks JJ, Higgins JPT, Altman DG (editors). Chapter 10: Analysing data and undertaking meta-analyses. In: Higgins JPT, Thomas J, Chandler J, Cumpston M, Li T, Page MJ, Welch VA (editors). Cochrane Handbook for Systematic Reviews of Interventions version 6.3 (updated February 2022). Cochrane, 2022. Available from www.training.cochrane.org/handbook.

REVIEWER #2

Reviewer #2: Some final comments I would like to see addressed before publication:

Introduction, paragraph 2: please make it clear here that you are focusing on severe falciparum malaria. Falciparum and knowlesi are the two parasites which sequester in the microvasculature thus causing severe disease. Severe vivax is very different in terms of presentation and the exact definition is debated.

In addition severe falciparum is diagnosed from evidence of *asexual* stages in circulation.

Response: Thank you. We have added a statement to indicate this. (Line 65-68) 

“Malaria is febrile illness that is spread through the infected bites from female Anopheles mosquitos and it is caused by the Plasmodium spp (P.malariae, P.falciparum, P.vivax, P.knowlesi, P.ovale and P.cynomolgi), of which P.falciparum is the major cause of severe illness, therefore, severe P.falciparum malaria is the focus of this study [3].”

Results: surely Figure 2 is wrong? In panel A the circle for ASU (artesunate) is very small, whereas ~3000 patients were randomised to ASU? Maybe I am missing something, but I don't quite understand this Figure.

Response: Thank you. Response: Thank you. Apologies. The weighting for the nodes was incorrect. This has been corrected. The size of the blue nodes are now proportional to the number of participants allocated to that drug.

Discussion: the authors write: “All artemisinin drugs except artemether may increase neurological sequela events in the first three to seven days of treatment, compared to quinine. Most of the neurological sequela events resolved during follow up, which is similar to findings of other studies [5,40,53].”

This should be caveated by the bias I mentioned in my first review: death and coma recovery are competing events: a drug that saves lives in patients who would have died with a different drug could thus result in longer coma recovery times (those patients would have died!). The authors do not have access to IPD and thus the effect estimates are likely to be biased.

Response: Thank you. We acknowledge that this is an important addition to the limitations of this study so we have included this in the Discussion. (Line 413, 425-427)

“However, we observed that mortality is a competing event with these adverse outcomes.” 

“Mortality may be considered as a competing event with some of the secondary outcomes but we were unable to explore this using more appropriate methods because individual participant data was not available.”

In response to reviewer comments, the authors wrote:

“We have adjusted our judgements on within-study-bias in the CINeMA webpage to accommodate the Reviewers concerns and have ammended this in S14. This is seen in the Contribution Matrix presented below.”

However when I look at the Sup Material page 5, I still see a yellow "!" for Hien 1996 and Van Hensbroek 1996. Have the authors changed their assessment of “concerns about the randomization process” for these two studies? I strongly believe that the randomization process was more than adequate in these two studies as explained in detail in the first review.

Response: Thank you. We apologise for the omission. The initial change after the first review was made in the quality of evidence assessment, we have now effected this change on the Risk of Bias Chart (S4) and the manuscript (Line 244).

The authors wrote:

“This is one of the main findings of this review. In this review, we did not systematically look at pharmacokinetics or studies detailing biological mechanisms, instead we looked at randomized controlled trials, which provide the best evidence for determining treatments effectiveness.”

For death - the main outcome we care about in the treatment of severe malaria (a disease with 10-30% mortality!) - it is pretty clear that artesunate is better than quinine. I agree that RCTs are the best evidence here.

For parasite clearance, this is also very clear: this is the main surrogate marker of drug activity in vivo.

Neurological sequelae have not be very well documented in these trials as it is very hard to consistently follow patients up after hospital discharge (Hien 1996 is only large trial to have done proper follow-up). As I have stated before, coma recovery times based on aggregated data are biased. So, no, I do not think that a main finding is a lack of consistency. A main finding is lack of data or lack of access to IPD for these outcomes. IPD meta-analysis would be much better to estimate the effects for these secondary outcomes.

Response: Thank you. We acknowledge that this is an important addition to the limitations of this study so we have added a statement in the Discussion to reflect this. (Line 425-427) 

“Mortality may be considered as a competing event with some of the secondary outcomes but we were unable to explore this using more appropriate methods because individual participant data was not available.”

---

## [Decision Letter · Decision Letter 2]

20 May 2022

Comparative efficacy and safety of the Artemisinin Derivatives compared to Quinine for treating Severe Malaria in Children and Adults: a Systematic Update of Literature and Network Meta-analysis

PONE-D-21-29782R2

Dear Dr. Nyaaba,

We’re pleased to inform you that your manuscript has been judged scientifically suitable for publication and will be formally accepted for publication once it meets all outstanding technical requirements.

Kind regards,

Benedikt Ley, PhD

Academic Editor

PLOS ONE

Reviewers' comments:

Reviewer's Responses to Questions

**Comments to the Author**

1. If the authors have adequately addressed your comments raised in a previous round of review and you feel that this manuscript is now acceptable for publication, you may indicate that here to bypass the “Comments to the Author” section, enter your conflict of interest statement in the “Confidential to Editor” section, and submit your "Accept" recommendation.

Reviewer #1: All comments have been addressed

Reviewer #2: All comments have been addressed

2. Is the manuscript technically sound, and do the data support the conclusions?

Reviewer #1: Yes

Reviewer #2: Yes

3. Has the statistical analysis been performed appropriately and rigorously? 

Reviewer #1: Yes

Reviewer #2: Yes

4. Have the authors made all data underlying the findings in their manuscript fully available?

Reviewer #1: Yes

Reviewer #2: (No Response)

5. Is the manuscript presented in an intelligible fashion and written in standard English?

Reviewer #1: Yes

Reviewer #2: Yes

6. Review Comments to the Author

Reviewer #1: The authors have adequately responded to the comments raised by the reviewers. I do not have any additional comments.

Reviewer #2: The authors have satisfactorily responded to all the comments. This paper provides a useful resource for evaluating the evidence base for the life saving benefit of the artemisinin derivatives in severe malaria.

7. PLOS authors have the option to publish the peer review history of their article (what does this mean?). If published, this will include your full peer review and any attached files.

Reviewer #1: No

Reviewer #2: **Yes: **James Watson

---

## [Editor Report · Acceptance letter]

12 Jul 2022

PONE-D-21-29782R2 

Comparative efficacy and safety of the Artemisinin Derivatives compared to Quinine for treating Severe Malaria in Children and Adults: a Systematic Update of Literature and Network Meta-analysis 

Dear Dr. Nyaaba:

I'm pleased to inform you that your manuscript has been deemed suitable for publication in PLOS ONE. Congratulations! Your manuscript is now with our production department. 

Kind regards, 

on behalf of

Dr Benedikt Ley 

Academic Editor

PLOS ONE